# HYPO: Hyperspherical Out-of-Distribution Generalization

**Haoyue Bai**[1,*] **Yifei Ming**[1,*] **Julian Katz-Samuels**[2,†] **Yixuan Li**[1]
Department of Computer Sciences, University of Wisconsin-Madison[1]
Amazon[2]
{baihaoyue,alvinming,sharonli}@cs.wisc.edu, jkatzsamuels@gmail.com

## Abstract

Out-of-distribution (OOD) generalization is critical for machine learning models deployed in the real world. However, achieving this can be fundamentally challenging, as it requires the ability to learn invariant features across different domains or environments. In this paper, we propose a novel framework HYPO (**HYP**erspherical **O**OD generalization) that provably learns domain-invariant representations in a hyperspherical space. In particular, our hyperspherical learning algorithm is guided by intra-class variation and inter-class separation principles—ensuring that features from the same class (across different training domains) are closely aligned with their class prototypes, while different class prototypes are maximally separated. We further provide theoretical justifications on how our prototypical learning objective improves the OOD generalization bound. Through extensive experiments on challenging OOD benchmarks, we demonstrate that our approach outperforms competitive baselines and achieves superior performance. Code is available at https://github.com/deeplearning-wisc/hypo.

## 1 Introduction

Deploying machine learning models in real-world settings presents a critical challenge of generalizing under distributional shifts. These shifts are common due to mismatches between the training and test data distributions. For instance, in autonomous driving, a model trained on in-distribution (ID) data collected under sunny weather conditions is expected to perform well in out-of-distribution (OOD) scenarios, such as rain or snow. This underscores the importance of the OOD generalization problem, which involves learning a predictor that can generalize across all possible environments, despite being trained on a finite subset of training environments.

A plethora of OOD generalization algorithms has been developed in recent years (Zhou et al., 2022), where a central theme is to learn domain-invariant representations—features that are consistent and meaningful across different environments (domains) and can generalize to the unseen test environment. Recently, Ye et al. (2021) theoretically showed that the OOD generalization error can be bounded in terms of intra-class *variation* and inter-class *separation*. Intra-class variation measures the stability of representations across different environments, while inter-class separation assesses the dispersion of features among different classes. Ideally, features should display low variation and high separation, in order to generalize well to OOD data (formally described in Section 3). Despite the theoretical analysis, a research question remains open in the field:

> **RQ:** How to design a practical learning algorithm that directly achieves these two properties, and what theoretical guarantees can the algorithm offer?

To address the question, this paper presents a learning framework HYPO (**HYP**erspherical **O**OD generalization), which provably learns domain-invariant representations in the hyperspherical space with unit norm (Section 4). Our key idea is to promote low variation (aligning representation across

---

*Equal contribution. Correspondence to Yifei Ming and Yixuan Li
†This work is not related to the author's position at Amazon.

domains for every class) and high separation (separating prototypes across different classes). In particular, the learning objective shapes the embeddings such that samples from the same class (across all training environments) gravitate towards their corresponding class prototype, while different class prototypes are maximally separated. The two losses in our objective function can be viewed as optimizing the key properties of intra-class variation and inter-class separation, respectively. Since samples are encouraged to have a small distance with respect to their class prototypes, the resulting embedding geometry can have a small distribution discrepancy across domains and benefits OOD generalization. Geometrically, we show that our loss function can be understood through the lens of maximum likelihood estimation under the classic von Mises-Fisher distribution.

**Empirical contribution.** Empirically, we demonstrate strong OOD generalization performance by extensively evaluating HYPO on common benchmarks (Section 5). On the CIFAR-10 (ID) vs. CIFAR-10-Corruption (OOD) task, HYPO substantially improves the OOD generalization accuracy on challenging cases such as Gaussian noise, from $78.09\%$ to $85.21\%$. Furthermore, we establish superior performance on popular domain generalization benchmarks, including PACS, Office-Home, VLCS, etc. For example, we achieve $88.0\%$ accuracy on PACS which outperforms the best loss-based method by $1.1\%$. This improvement is non-trivial using standard stochastic gradient descent optimization. When coupling our loss with specialized optimization SWAD (Cha et al., 2021), the accuracy is further increased to $89\%$. We provide visualization and quantitative analysis to verify that features learned by HYPO indeed achieve low intra-class variation and high inter-class separation.

**Theoretical insight.** We provide theoretical justification for how HYPO can guarantee improved OOD generalization, supporting our empirical findings. Our theory complements Ye et al. (2021), which does not provide a loss for optimizing the intra-class variation or inter-class separation. Thus, *a key contribution of this paper is to provide a crucial link between provable understanding and a practical algorithm for OOD generalization in the hypersphere.* In particular, our Theorem 6.1 shows that when the model is trained with our loss function, we can upper bound intra-class variation, a key quantity to bound OOD generalization error. For a learnable OOD generalization task, the upper bound on generalization error is determined by the variation estimate on the training environments, which is effectively reduced by our loss function under sufficient sample size and expressiveness of the neural network.

## 2 PROBLEM SETUP

We consider a multi-class classification task that involves a pair of random variables $(X, Y)$ over instances $\mathbf{x} \in \mathcal{X} \subset \mathbb{R}^d$ and corresponding labels $y \in \mathcal{Y} := \{1, 2, \cdots, C\}$. The joint distribution of $X$ and $Y$ is unknown and represented by $\mathbb{P}_{XY}$. The goal is to learn a predictor function, $f : \mathcal{X} \to \mathbb{R}^C$, that can accurately predict the label $y$ for an input $\mathbf{x}$, where $(\mathbf{x}, y) \sim \mathbb{P}_{XY}$.

Unlike in standard supervised learning tasks, the out-of-distribution (OOD) generalization problem is challenged by the fact that one cannot sample directly from $\mathbb{P}_{XY}$. Instead, we can only sample $(X, Y)$ under limited environmental conditions, each of which corrupts or varies the data differently. For example, in autonomous driving, these environmental conditions may represent different weathering conditions such as snow, rain, etc. We formalize this notion of environmental variations with a set of *environments* or domains $\mathcal{E}_{\text{all}}$. Sample pairs $(X^e, Y^e)$ are randomly drawn from environment $e$. In practice, we may only have samples from a finite subset of *available environments* $\mathcal{E}_{\text{avail}} \subset \mathcal{E}_{\text{all}}$. Given $\mathcal{E}_{\text{avail}}$, the goal is to learn a predictor $f$ that can generalize across all possible environments. The problem is stated formally below.

**Definition 2.1** (OOD Generalization). *Let $\mathcal{E}_{\text{avail}} \subset \mathcal{E}_{\text{all}}$ be a set of training environments, and assume that for each environment $e \in \mathcal{E}_{\text{avail}}$, we have a dataset $\mathcal{D}^e = \{(\mathbf{x}_j^e, y_j^e)\}_{j=1}^{n_e}$, sampled i.i.d. from an unknown distribution $\mathbb{P}_{XY}^e$. The goal of OOD generalization is to find a classifier $f^*$, using the data from the datasets $\mathcal{D}^e$, that minimizes the worst-case risk over the entire family of environments $\mathcal{E}_{\text{all}}$:*

$$\min_{f \in \mathcal{F}} \max_{e \in \mathcal{E}_{\text{all}}} \mathbb{E}_{\mathbb{P}_{XY}^e} \ell(f(X^e), Y^e), \tag{1}$$

*where $\mathcal{F}$ is hypothesis space and $l(\cdot, \cdot)$ is the loss function.*

The problem is challenging since we do not have access to data from domains outside $\mathcal{E}_{\text{avail}}$. In particular, the task is commonly referred to as multi-source domain generalization when $|\mathcal{E}_{\text{avail}}| > 1$.

## 3 MOTIVATION OF ALGORITHM DESIGN

Our work is motivated by the theoretical findings in Ye et al. (2021), which shows that the OOD generalization performance can be bounded in terms of intra-class *variation* and inter-class *separation* with respect to various environments. The formal definitions are given as follows.

**Definition 3.1** (Intra-class variation)**.** *The variation of feature $\phi$ across a domain set $\mathcal{E}$ is*

$$\mathcal{V}(\phi, \mathcal{E}) = \max_{y \in \mathcal{Y}} \sup_{e,e' \in \mathcal{E}} \rho\big(\mathbb{P}(\phi^e|y), \mathbb{P}(\phi^{e'}|y)\big), \tag{2}$$

*where $\rho(\mathbb{P}, \mathbb{Q})$ is a symmetric distance (e.g., Wasserstein distance, total variation, Hellinger distance) between two distributions, and $\mathbb{P}(\phi^e|y)$ denotes the class-conditional distribution for features of samples in environment $e$.*

**Definition 3.2** (Inter-class separation[1])**.** *The separation of feature $\phi$ across domain set $\mathcal{E}$ is*

$$\mathcal{I}_\rho(\phi, \mathcal{E}) = \frac{1}{C(C-1)} \sum_{\substack{y \neq y' \\ y,y' \in \mathcal{Y}}} \min_{e \in \mathcal{E}} \rho\big(\mathbb{P}(\phi^e|y), \mathbb{P}(\phi^e|y')\big). \tag{3}$$

The intra-class variation $\mathcal{V}(\phi, \mathcal{E})$ measures the stability of feature $\phi$ over the domains in $\mathcal{E}$ and the inter-class separation $\mathcal{I}(\phi, \mathcal{E})$ captures the ability of $\phi$ to distinguish different labels. Ideally, features should display high separation and low variation.

**Definition 3.3.** *The OOD generalization error of classifier $f$ is defined as follows:*

$$err(f) = \max_{e \in \mathcal{E}_{all}} \mathbb{E}_{\mathbb{P}^e_{XY}} \ell(f(X^e), Y^e) - \max_{e \in \mathcal{E}_{avail}} \mathbb{E}_{\mathbb{P}^e_{XY}} \ell(f(X^e), Y^e)$$

*which is bounded by the variation estimate on $\mathcal{E}_{avail}$ with the following theorem.*

**Theorem 3.1** (OOD error upper bound, informal (Ye et al., 2021))**.** *Suppose the loss function $\ell(\cdot, \cdot)$ is bounded by $[0, B]$. For a learnable OOD generalization problem with sufficient inter-class separation, the OOD generalization error $err(f)$ can be upper bounded by*

$$err(f) \leq O\Big(\big(\mathcal{V}^{\text{sup}}(h, \mathcal{E}_{\text{avail}})\big)^{\frac{\alpha^2}{(\alpha+d)^2}}\Big), \tag{4}$$

*for some $\alpha > 0$, and $\mathcal{V}^{\text{sup}}(h, \mathcal{E}_{\text{avail}}) \triangleq \sup_{\beta \in \mathcal{S}^{d-1}} \mathcal{V}(\beta^\top h, \mathcal{E}_{\text{avail}})$ is the inter-class variation, $h(\cdot) \in \mathbb{R}^d$ is the feature vector, and $\beta$ is a vector in unit hypersphere $\mathcal{S}^{d-1} = \{\beta \in \mathbb{R}^d : \|\beta\|_2 = 1\}$, and $f$ is a classifier based on normalized feature $h$.*

**Remarks.** The Theorem above suggests that both low intra-class variation and high inter-class separation are desirable properties for theoretically grounded OOD generalization. Note that in the **full formal** Theorem (see Appendix C), maintaining the inter-class separation is necessary for the learnability of the OOD generalization problem (Def. C.2). In other words, when the learned embeddings exhibit high inter-class separation, the problem becomes learnable. In this context, bounding intra-class variation becomes crucial for reducing the OOD generalization error.

Despite the theoretical underpinnings, it remains unknown to the field how to design a practical learning algorithm that directly achieves these two properties, and what theoretical guarantees can the algorithm offer. This motivates our work.

> To reduce the OOD generalization error, our key motivation is to design a hyperspherical learning algorithm that directly promotes low variation (aligning representation across domains for every class) and high separation (separating prototypes across different classes).

## 4 METHOD

Following the motivation in Section 3, we now introduce the details of the learning algorithm HYPO (**HYP**erspherical **O**OD generalization), which is designed to promote domain invariant representations

---

[1]Referred to as "Informativeness" in Ye et al. (2021).

in the hyperspherical space. The key idea is to shape the hyperspherical embedding space so that samples from the same class (across all training environments $\mathcal{E}_{\text{avail}}$) are closely aligned with the corresponding class prototype. Since all points are encouraged to have a small distance with respect to the class prototypes, the resulting embedding geometry can have a small distribution discrepancy across domains and hence benefits OOD generalization. In what follows, we first introduce the learning objective (Section 4.1), and then we discuss the geometrical interpretation of the loss and embedding (Section 4.2). We will provide theoretical justification for HYPO in Section 6, which leads to a provably smaller intra-class variation, a key quantity to bound OOD generalization error.

## 4.1 Hyperspherical Learning for OOD Generalization

**Loss function.** The learning algorithm is motivated to directly optimize the two criteria: intra-class variation and inter-class separation. At a high level, HYPO aims to learn embeddings for each sample in the training environments by maintaining a class prototype vector $\boldsymbol{\mu}_c \in \mathbb{R}^d$ for each class $c \in \{1, 2, ..., C\}$. To optimize for low variation, the loss encourages the feature embedding of a sample to be close to its class prototype. To optimize for high separation, the loss encourages different class prototypes to be far apart from each other.

Specifically, we consider a deep neural network $h : \mathcal{X} \mapsto \mathbb{R}^d$ that maps an input $\tilde{\mathbf{x}} \in \mathcal{X}$ to a feature embedding $\tilde{\mathbf{z}} := h(\tilde{\mathbf{x}})$. The loss operates on the normalized feature embedding $\mathbf{z} := \tilde{\mathbf{z}}/\|\tilde{\mathbf{z}}\|_2$. The normalized embeddings are also referred to as *hyperspherical embeddings*, since they are on a unit hypersphere, denoted as $S^{d-1} := \{\mathbf{z} \in \mathbb{R}^d \mid \|\mathbf{z}\|_2 = 1\}$. The loss is formalized as follows:

$$\mathcal{L} = -\underbrace{\frac{1}{N} \sum_{e \in \mathcal{E}_{\text{avail}}} \sum_{i=1}^{|\mathcal{D}^e|} \log \frac{\exp\left(\mathbf{z}_i^{e\top} \boldsymbol{\mu}_{c(i)}/\tau\right)}{\sum_{j=1}^{C} \exp\left(\mathbf{z}_i^{e\top} \boldsymbol{\mu}_j/\tau\right)}}_{\mathcal{L}_{\text{var}}: \downarrow \textbf{variation}} + \underbrace{\frac{1}{C} \sum_{i=1}^{C} \log \frac{1}{C-1} \sum_{j \neq i, j \in \mathcal{Y}} \exp\left(\boldsymbol{\mu}_i^\top \boldsymbol{\mu}_j/\tau\right)}_{\uparrow \textbf{separation}},$$

where $N$ is the number of samples, $\tau$ is the temperature, $\mathbf{z}$ is the normalized feature embedding, and $\boldsymbol{\mu}_c$ is the prototype embedding for class $c$. While hyperspherical learning algorithms have been studied in other context (Mettes et al., 2019; Khosla et al., 2020; Ming et al., 2023), *none of the prior works explored its provable connection to domain generalization, which is our distinct contribution. We will theoretically show in Section 6 that minimizing our loss function effectively reduces intra-class variation, a key quantity to bound OOD generalization error.*

The training objective in Equation 5 can be efficiently optimized end-to-end. During training, an important step is to estimate the class prototype $\boldsymbol{\mu}_c$ for each class $c \in \{1, 2, ..., C\}$. The class-conditional prototypes can be updated in an exponential-moving-average manner (EMA) (Li et al., 2020):

$$\boldsymbol{\mu}_c := \text{Normalize}(\alpha\boldsymbol{\mu}_c + (1-\alpha)\mathbf{z}), \ \forall c \in \{1, 2, \ldots, C\} \tag{5}$$

where the prototype $\boldsymbol{\mu}_c$ for class $c$ is updated during training as the moving average of all embeddings with label $c$, and $\mathbf{z}$ denotes the normalized embedding of samples of class $c$. An end-to-end pseudo algorithm is summarized in Appendix A.

**Class prediction.** In testing, classification is conducted by identifying the closest class prototype: $\hat{y} = \arg\max_{c \in [C]} f_c(\mathbf{x})$, where $f_c(\mathbf{x}) = \mathbf{z}^\top \boldsymbol{\mu}_c$ and $\mathbf{z} = \frac{h(\mathbf{x})}{\|h(\mathbf{x})\|_2}$ is the normalized feature embedding.

## 4.2 Geometrical Interpretation of Loss and Embedding

Geometrically, the loss function above can be interpreted as learning embeddings located on the surface of a unit hypersphere. The hyperspherical embeddings can be modeled by the von Mises-Fisher (vMF) distribution, a well-known distribution in directional statistics (Jupp & Mardia, 2009). For a unit vector $\mathbf{z} \in \mathbb{R}^d$ in class $c$, the probability density function is defined as

$$p(\mathbf{z} \mid y = c) = Z_d(\kappa) \exp(\kappa \boldsymbol{\mu}_c^\top \mathbf{z}), \tag{6}$$

where $\boldsymbol{\mu}_c \in \mathbb{R}^d$ denotes the mean direction of the class $c$, $\kappa \geq 0$ denotes the concentration of the distribution around $\boldsymbol{\mu}_c$, and $Z_d(\kappa)$ denotes the normalization factor. A larger $\kappa$ indicates a higher concentration around the class center. In the extreme case of $\kappa = 0$, the samples are distributed uniformly on the hypersphere.

Under this probabilistic model, an embedding $\mathbf{z}$ is assigned to the class $c$ with the following probability

$$p(y = c \mid \mathbf{z}; \{\kappa, \boldsymbol{\mu}_j\}_{j=1}^C) = \frac{Z_d(\kappa) \exp(\kappa \boldsymbol{\mu}_c^\top \mathbf{z})}{\sum_{j=1}^C Z_d(\kappa) \exp(\kappa \boldsymbol{\mu}_j^\top \mathbf{z})}$$

$$= \frac{\exp(\boldsymbol{\mu}_c^\top \mathbf{z}/\tau)}{\sum_{j=1}^C \exp(\boldsymbol{\mu}_j^\top \mathbf{z}/\tau)}, \quad (7)$$

where $\tau = 1/\kappa$ denotes a temperature parameter.

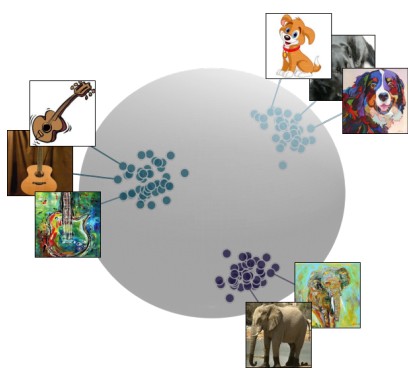

**Maximum likelihood view.** Notably, minimizing the first term in our loss (*cf.* Eq. 5) is equivalent to performing maximum likelihood estimation under the vMF distribution:

Figure 1: Illustration of hyperspherical embeddings. Images are from PACS (Li et al., 2017).

$$\mathrm{argmax}_\theta \prod_{i=1}^N p(y_i \mid \mathbf{x}_i; \{\kappa, \boldsymbol{\mu}_j\}_{j=1}^C), \text{ where } (\mathbf{x}_i, y_i) \in \bigcup_{e \in \mathcal{E}_{\text{train}}} \mathcal{D}^e$$

where $i$ is the index of sample, $j$ is the index of the class, and $N$ is the size of the training set. In effect, this loss encourages each ID sample to have a high probability assigned to the correct class in the mixtures of the vMF distributions.

## 5 EXPERIMENTS

In this section, we show that HYPO achieves strong OOD generalization performance in practice, establishing competitive performance on several benchmarks. In what follows, we describe the experimental setup in Section 5.1, followed by main results and analysis in Section 5.2.

### 5.1 EXPERIMENTAL SETUP

**Datasets.** Following the common benchmarks in literature, we use CIFAR-10 (Krizhevsky et al., 2009) as the in-distribution data. We use CIFAR-10-C (Hendrycks & Dietterich, 2019) as OOD data, with 19 different common corruption applied to CIFAR-10. In addition to CIFAR-10, we conduct experiments on popular benchmarks including PACS (Li et al., 2017), Office-Home (Gulrajani & Lopez-Paz, 2020), and VLCS (Gulrajani & Lopez-Paz, 2020) to validate the generalization performance. PACS contains 4 domains/environments (photo, art painting, cartoon, sketch) with 7 classes (dog, elephant, giraffe, guitar, horse, house, person). Office-Home comprises four different domains: art, clipart, product, and real. Results on additional OOD datasets Terra Incognita (Gulrajani & Lopez-Paz, 2020), and ImageNet can be found in Appendix F and Appendix G.

**Evaluation metrics.** We report the following two metrics: **(1)** ID classification accuracy (ID Acc.) for ID generalization, and **(2)** OOD classification accuracy (OOD Acc.) for OOD generalization.

**Experimental details.** In our main experiments, we use ResNet-18 for CIFAR-10 and ResNet-50 for PACS, Office-Home, and VLCS. For these datasets, we use stochastic gradient descent with momentum 0.9, and weight decay $10^{-4}$. For CIFAR-10, we train the model from scratch for 500 epochs using an initial learning rate of 0.5 and cosine scheduling, with a batch size of 512. Following common practice for contrastive losses (Chen et al., 2020; Khosla et al., 2020; Yao et al., 2022), we use an MLP projection head with one hidden layer to obtain features. The embedding (output) dimension is 128 for the projection head. We set the default temperature $\tau$ as 0.1 and the prototype update factor $\alpha$ as 0.95. For PACS, Office-Home, and VLCS, we follow the common practice and initialize the network using ImageNet pre-trained weights. We fine-tune the network for 50 epochs. The embedding dimension is 512 for the projection head. We adopt the leave-one-domain-out evaluation protocol and use the training domain validation set for model selection (Gulrajani & Lopez-Paz, 2020), where the validation set is pooled from all training domains. Details on other hyperparameters are in Appendix D.

| Algorithm | PACS | Office-Home | VLCS | Average Acc. (%) |
|---|---|---|---|---|
| **ERM** (Vapnik, 1999) | 85.5 | 67.6 | 77.5 | 76.7 |
| **CORAL** (Sun & Saenko, 2016) | 86.2 | 68.7 | 78.8 | 77.9 |
| **DANN** (Ganin et al., 2016) | 83.7 | 65.9 | 78.6 | 76.1 |
| **MLDG** (Li et al., 2018a) | 84.9 | 66.8 | 77.2 | 76.3 |
| **CDANN** (Li et al., 2018c) | 82.6 | 65.7 | 77.5 | 75.3 |
| **MMD** (Li et al., 2018b) | 84.7 | 66.4 | 77.5 | 76.2 |
| **IRM** (Arjovsky et al., 2019) | 83.5 | 64.3 | 78.6 | 75.5 |
| **GroupDRO** (Sagawa et al., 2020) | 84.4 | 66.0 | 76.7 | 75.7 |
| **I-Mixup** (Wang et al., 2020; Xu et al., 2020; Yan et al., 2020) | 84.6 | 68.1 | 77.4 | 76.7 |
| **RSC** (Huang et al., 2020) | 85.2 | 65.5 | 77.1 | 75.9 |
| **ARM** (Zhang et al., 2021) | 85.1 | 64.8 | 77.6 | 75.8 |
| **MTL** (Blanchard et al., 2021) | 84.6 | 66.4 | 77.2 | 76.1 |
| **VREx** (Krueger et al., 2021) | 84.9 | 66.4 | 78.3 | 76.5 |
| **Mixstyle** (Zhou et al., 2021) | 85.2 | 60.4 | 77.9 | 74.5 |
| **SelfReg** (Kim et al., 2021) | 85.6 | 67.9 | 77.8 | 77.1 |
| **SagNet** (Nam et al., 2021) | 86.3 | 68.1 | 77.8 | 77.4 |
| **GVRT** (Min et al., 2022) | 85.1 | 70.1 | 79.0 | 78.1 |
| **VNE** (Kim et al., 2023) | 86.9 | 65.9 | 78.1 | 77.0 |
| **HYPO (Ours)** | $88.0_{\pm 0.4}$ | $71.7_{\pm 0.3}$ | $78.2_{\pm 0.4}$ | **79.3** |

Table 1: Comparison with domain generalization methods on the PACS, Office-Home, and VLCS. All methods are trained on ResNet-50. The model selection is based on a training domain validation set. To isolate the effect of loss functions, all methods are optimized using standard SGD. We report the average and std of our method. $\pm x$ denotes the rounded standard error.

## 5.2 MAIN RESULTS AND ANALYSIS

**HYPO excels on common corruption benchmarks.** As shown in Figure 2, HYPO achieves consistent improvement over the ERM baseline (trained with cross-entropy loss), on a variety of common corruptions. Our evaluation includes different corruptions including Gaussian noise, Snow, JPEG compression, Shot noise, Zoom blur, etc. The model is trained on CIFAR-10, without seeing any type of corruption data. In particular, our method brings significant improvement for challenging cases such as Gaussian noise, enhancing OOD accuracy from 78.09% to 85.21% (**+7.12**%). Complete results on all 19 different corruption types are in Appendix E.

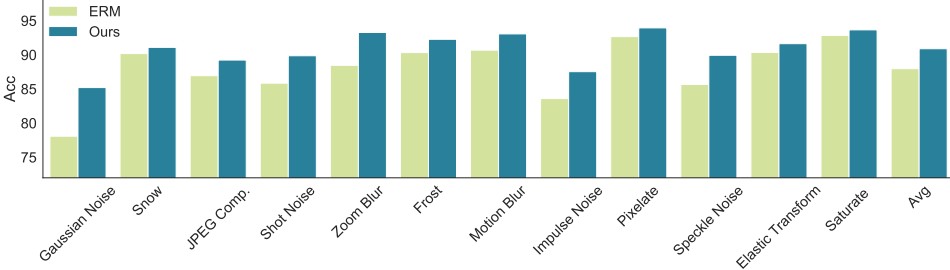

Figure 2: Our method HYPO significantly improves the OOD generalization performance compared to ERM on various OOD datasets w.r.t. CIFAR-10 (ID). Full results can be seen in Appendix E.

**HYPO establishes competitive performance on popular benchmarks.** Our method delivers superior results in the popular domain generalization tasks, as shown in Table 1. HYPO outperforms an extensive collection of common OOD generalization baselines on popular domain generalization datasets, including PACS, Office-Home, VLCS. For instance, on PACS, HYPO improves the best loss-based method by **1.1**%. Notably, this enhancement is non-trivial since we are not relying on specialized optimization algorithms such as SWAD (Cha et al., 2021). Later in our ablation, we show that coupling HYPO with SWAD can further boost the OOD generalization performance, establishing superior performance on this challenging task.

With multiple training domains, we observe that it is desirable to emphasize hard negative pairs when optimizing the inter-class separation. As depicted in Figure 3, the embeddings of negative pairs from the same domain but different classes (such as dog and elephant in art painting) can be quite close on the hypersphere. Therefore, it is more informative to separate such hard negative pairs. This can be enforced by a simple modification to the denominator of our variation loss (Eq. 11 in Appendix D), which we adopt for multi-source domain generalization tasks.

| Algorithm | Art painting | Cartoon | Photo | Sketch | Average Acc. (%) |
|---|---|---|---|---|---|
| **PCL w/ SGD** (Yao et al., 2022) | 88.0 | 78.8 | 98.1 | 80.3 | 86.3 |
| **HYPO w/ SGD (Ours)** | 87.2 | 82.3 | 98.0 | 84.5 | **88.0** |
| **PCL w/ SWAD** (Yao et al., 2022) | 90.2 | 83.9 | 98.1 | 82.6 | 88.7 |
| **HYPO w/ SWAD (Ours)** | 90.5 | 84.6 | 97.7 | 83.2 | **89.0** |

Table 2: Results for comparing PCL and HYPO with SGD-based and SWAD-based optimizations on the PACS benchmark. (*The performance reported in the original PCL paper Table 3 is implicitly based on SWAD).

**Relations to PCL.** PCL (Yao et al., 2022) adapts a proxy-based contrastive learning framework for domain generalization. We highlight several notable distinctions from ours: **(1)** While PCL offers no theoretical insights, HYPO is guided by theory. We provide a formal theoretical justification that our method reduces intra-class variation which is essential to bounding OOD generalization error (see Section 6); **(2)** Our loss function formulation is different and can be rigorously interpreted as shaping vMF distributions of hyperspherical embeddings (see Section 4.2), whereas PCL can not; **(3)** Unlike PCL (86.3% w/o SWAD), HYPO is able to achieve competitive performance (88.0%) without heavy reliance on special optimization SWAD (Cha et al., 2021), a dense and overfit-aware stochastic weight sampling (Izmailov et al., 2018) strategy for OOD generalization. As shown in Table 2, we also conduct experiments in conjunction with SWAD. Compared to PCL, HYPO achieves superior performance with **89**% accuracy, which further demonstrates its advantage.

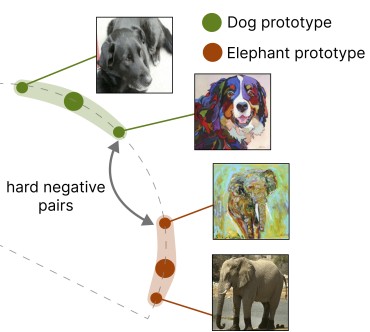

Figure 3: Illustration of hard negative pairs which share the same domain (art painting) but have different class labels.

**Visualization of embedding.** Figure 4 shows the UMAP (McInnes et al., 2018) visualization of feature embeddings for ERM (left) vs. HYPO (right). The embeddings are extracted from models trained on PACS. The red, orange, and green points are from the in-distribution, corresponding to art painting (A), photo (P), and sketch (S) domains. The violet points are from the unseen OOD domain cartoon (C). There are two salient observations: **(1)** for any given class, the embeddings across domains $\mathcal{E}_{\text{all}}$ become significantly more aligned (and invariant) using our method compared to the ERM baseline. This directly verifies the low variation (*cf.* Equation 2) of our learned embedding. **(2)** The embeddings are well separated across different classes, and distributed more uniformly in the space than ERM, which verifies the high inter-class separation (*cf.* Equation 3) of our method. Overall, our observations well support the efficacy of HYPO.

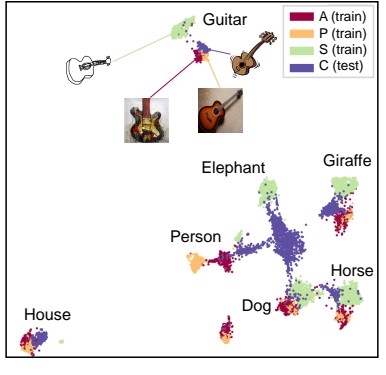

(a) ERM (**high variation**)

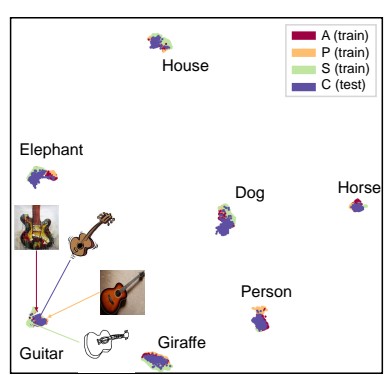

(b) HYPO (**low variation**)

Figure 4: UMAP (McInnes et al., 2018) visualization of the features when the model is trained with CE vs. HYPO for PACS. The red, orange, and green points are from the in-distribution, which denote art painting (A), photo (P), and sketch (S). The violet points are from the unseen OOD domain cartoon (C).

**Quantitative verification of intra-class variation.** We provide empirical verification on intra-class variation in Figure 5, where the model is trained on PACS. We measure the intra-class *variation* with Sinkhorn divergence (entropy regularized Wasserstein distance). The horizontal axis (0)-(6) denotes

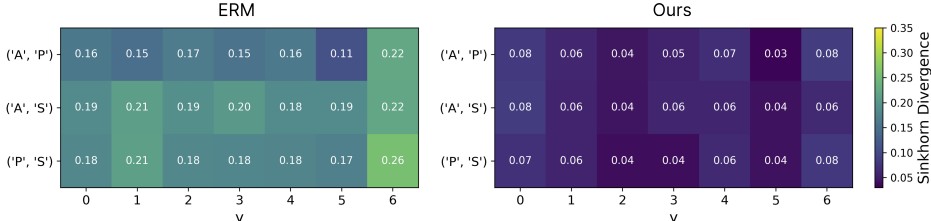

Figure 5: Intra-class variation for ERM (left) vs. HYPO (right) on PACS. For each class $y$, we measure the Sinkhorn Divergence between the embeddings of each pair of domains. Our method results in significantly lower intra-class variation across different pairs of training domains compared to ERM.

different classes, and the vertical axis denotes different pairs of training domains ('P', 'A', 'S'). Darker color indicates lower Sinkhorn divergence. We can see that our method results in significantly lower intra-class variation compared to ERM, which aligns with our theoretical insights in Section 6.

**Additional ablation studies.** *Due to space constraints, we defer additional experiments and ablations to the Appendix, including **(1)** results on other tasks from DomainBed (Appendix F); **(2)** results on large-scale benchmarks such as ImageNet-100 (Appendix G); **(3)** ablation of different loss terms (Appendix H); **(4)** an analysis on the effect of $\tau$ and $\alpha$ (Appendix I).*

## 6    WHY HYPO IMPROVES OUT-OF-DISTRIBUTION GENERALIZATION?

In this section, we provide a formal justification of the loss function. Our main Theorem 6.1 gives a provable understanding of how the learning objective effectively reduces the variation estimate $\mathcal{V}^{\text{sup}}(h, \mathcal{E}_{\text{avail}})$, thus directly reducing the OOD generalization error according to Theorem 3.1. For simplicity, we assume $\tau = 1$ and denote the prototype vectors $\boldsymbol{\mu}_1, \ldots, \boldsymbol{\mu}_C \in \mathcal{S}^{d-1}$. Let $\mathcal{H} \subset \{h : \mathcal{X} \mapsto \mathcal{S}^{d-1}\}$ denote the function class induced by the neural network.

**Theorem 6.1** (Variation upper bound using HYPO). *When samples are aligned with class prototypes such that $\frac{1}{N} \sum_{j=1}^{N} \boldsymbol{\mu}_{c(j)}^{\top} \mathbf{z}_j \geq 1 - \epsilon$ for some $\epsilon \in (0, 1)$, then $\exists \delta \in (0, 1)$, with probability at least $1 - \delta$,*

$$\mathcal{V}^{\text{sup}}(h, \mathcal{E}_{\text{avail}}) \leq O\left(\epsilon^{1/3} + \left(\frac{\ln(2/\delta)}{N}\right)^{1/6} + \left(\mathbb{E}_{\mathcal{D}}\left[\frac{1}{N}\mathbb{E}_{\sigma_1,\ldots,\sigma_N} \sup_{h \in \mathcal{H}} \sum_{i=1}^{N} \sigma_i \mathbf{z}_i^{\top} \boldsymbol{\mu}_{c(i)}\right]\right)^{1/3}\right),$$

*where $\mathbf{z}_j = \frac{h(\mathbf{x}_j)}{\|h(\mathbf{x}_j)\|_2}$, $\sigma_1, \ldots, \sigma_N$ are Rademacher random variables and $O(\cdot)$ suppresses dependence on constants and $|\mathcal{E}_{\text{avail}}|$.*

**Implications.** In Theorem 6.1, we can see that the upper bound consists of three factors: the optimization error, the Rademacher complexity of the given neural network, and the estimation error which becomes close to 0 as the number of samples $N$ increases. Importantly, the term $\epsilon$ reflects how sample embeddings are aligned with their class prototypes on the hyperspherical space (as we have $\frac{1}{N} \sum_{j=1}^{N} \boldsymbol{\mu}_{c(j)}^{\top} \mathbf{z}_j \geq 1 - \epsilon$), *which is directly minimized by our proposed loss in Equation 5*. The above Theorem implies that when we train the model with the HYPO loss, we can effectively upper bound the intra-class variation, a key term for bounding OOD generation performance by Theorem 3.1. In Section H, we provide empirical verification of our bound by estimating $\hat{\epsilon}$, which is indeed close to 0 for models trained with HYPO loss. We defer proof details to Appendix C.

**Necessity of inter-class separation loss.** We further present a theoretical analysis in Appendix J explaining how our loss promotes inter-class separation, which is necessary to ensure the learnability of the OOD generalization problem. We provide a brief summary in Appendix C and discuss the notion of OOD learnability, and would like to refer readers to Ye et al. (2021) for an in-depth and formal treatment. Empirically, to verify the impact of inter-class separation, we conducted an ablation study in Appendix H, where we compare the OOD performance of our method (with separation loss) vs. our method (without separation loss). We observe that incorporating separation loss indeed achieves stronger OOD generalization performance, echoing the theory.

## 7    RELATED WORKS

**Out-of-distribution generalization.**    OOD generalization is an important problem when the training and test data are sampled from different distributions. Compared to domain adaptation (Daume III & Marcu, 2006; Ben-David et al., 2010; Tzeng et al., 2017; Kang et al., 2019; Wang et al., 2022c), OOD generalization is more challenging (Blanchard et al., 2011; Muandet et al., 2013; Gulrajani & Lopez-Paz, 2020; Bai et al., 2021b; Zhou et al., 2021; Koh et al., 2021; Bai et al., 2021a; Wang et al., 2022b; Ye et al., 2022; Cha et al., 2022; Bai et al., 2023; Kim et al., 2023; Guo et al., 2023; Dai et al., 2023; Tong et al., 2023), which aims to generalize to unseen distributions without any sample from the target domain. In particular, A popular direction is to extract domain-invariant feature representation. Prior works show that the invariant features from training domains can help discover invariance on target domains for linear models (Peters et al., 2016; Rojas-Carulla et al., 2018). IRM (Arjovsky et al., 2019) and its variants (Ahuja et al., 2020; Krueger et al., 2021) aim to find invariant representation from different training domains via an invariant risk regularizer. Mahajan et al. (2021) propose a causal matching-based algorithm for domain generalization. Other lines of works have explored the problem from various perspectives such as causal discovery (Chang et al., 2020), distributional robustness (Sagawa et al., 2020; Zhou et al., 2020), model ensembles (Chen et al., 2023b; Rame et al., 2023), and test-time adaptation (Park et al., 2023; Chen et al., 2023a). In this paper, we focus on improving OOD generalization via hyperspherical learning, and provide a new theoretical analysis of the generalization error.

**Theory for OOD generalization.**    Although the problem has attracted great interest, theoretical understanding of desirable conditions for OOD generalization is under-explored. Generalization to arbitrary OOD is impossible since the test distribution is unknown (Blanchard et al., 2011; Muandet et al., 2013). Numerous general distance measures exist for defining a set of test domains around the training domain, such as KL divergence (Joyce, 2011), MMD (Gretton et al., 2006), and EMD (Rubner et al., 1998). Based on these measures, some prior works focus on analyzing the OOD generalization error bound. For instance, Albuquerque et al. (2019) obtain a risk bound for linear combinations of training domains. Ye et al. (2021) provide OOD generalization error bounds based on the notation of variation. In this work, we provide a hyperspherical learning algorithm that provably reduces the variation, thereby improving OOD generalization both theoretically and empirically.

**Contrastive learning for domain generalization**    Contrastive learning methods have been widely explored in different learning tasks. For example, Wang & Isola (2020) analyze the relation between the alignment and uniformity properties on the hypersphere for unsupervised learning, while we focus on supervised learning with domain shift. Tapaswi et al. (2019) investigates a contrastive metric learning approach for hyperspherical embeddings in video face clustering, which differs from our objective of OOD generalization. Von Kügelgen et al. (2021) provide theoretical justification for self-supervised learning with data augmentations. Recently, contrastive losses have been adopted for OOD generalization. For example, CIGA (Chen et al., 2022) captures the invariance of graphs to enable OOD generalization for graph data. CNC (Zhang et al., 2022) is specifically designed for learning representations robust to spurious correlation by inferring pseudo-group labels and performing supervised contrastive learning. SelfReg (Kim et al., 2021) proposes a self-supervised contrastive regularization for domain generalization with non-hyperspherical embeddings, while we focus on hyperspherical features with theoretically grounded loss formulations.

## 8    CONCLUSION

In this paper, we present a theoretically justified algorithm for OOD generalization via hyperspherical learning. HYPO facilitates learning domain-invariant representations in the hyperspherical space. Specifically, we encourage low variation via aligning features across domains for each class and promote high separation by separating prototypes across different classes. Theoretically, we provide a provable understanding of how our loss function reduces the OOD generalization error. Minimizing our learning objective can reduce the variation estimates, which determine the general upper bound on the generalization error of a learnable OOD generalization task. Empirically, HYPO achieves superior performance compared to competitive OOD generalization baselines. We hope our work can inspire future research on OOD generalization and provable understanding.

ACKNOWLEDGEMENT

The authors would like to thank ICLR anonymous reviewers for their helpful feedback. The work is supported by the AFOSR Young Investigator Program under award number FA9550-23-1-0184, National Science Foundation (NSF) Award No. IIS-2237037 & IIS-2331669, and Office of Naval Research under grant number N00014-23-1-2643.

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

## A  PSEUDO ALGORITHM

The training scheme of HYPO is shown below. We jointly optimize for (1) *low variation*, by encouraging the feature embedding of samples to be close to their class prototypes; and (2) *high separation*, by encouraging different class prototypes to be far apart from each other.

---

**Algorithm 1:** Hyperspherical Out-of-Distribution Generalization

---

1  **Input:** Training dataset $\mathcal{D}$, deep neural network encoder $h$, class prototypes $\boldsymbol{\mu}_c$ ($1 \leq j \leq C$), temperature $\tau$

2  **for** $epoch = 1, 2, \ldots,$ **do**

3    **for** $iter = 1, 2, \ldots,$ **do**

4       sample a mini-batch $B = \{\mathbf{x}_i, y_i\}_{i=1}^b$

5       obtain augmented batch $\tilde{B} = \{\tilde{\mathbf{x}}_i, \tilde{y}_i\}_{i=1}^{2b}$ by applying two random augmentations to $\mathbf{x}_i \in B \; \forall i \in \{1, 2, \ldots, b\}$

6       **for** $\tilde{\mathbf{x}}_i \in \tilde{B}$ **do**

         `// obtain normalized embedding`

7          $\tilde{\mathbf{z}}_i = h(\tilde{\mathbf{x}}_i), \mathbf{z}_i = \tilde{\mathbf{z}}_i / \|\tilde{\mathbf{z}}_i\|_2$

         `// update class-prototypes`

8          $\boldsymbol{\mu}_c := \text{Normalize}(\alpha \boldsymbol{\mu}_c + (1 - \alpha) \mathbf{z}_i), \; \forall c \in \{1, 2, \ldots, C\}$

      `// calculate the loss for low variation`

9       $\mathcal{L}_{\text{var}} = -\frac{1}{N} \sum_{e \in \mathcal{E}_{\text{avail}}} \sum_{i=1}^{|\mathcal{D}^e|} \log \frac{\exp(\mathbf{z}_i^{e\top} \boldsymbol{\mu}_{c(i)} / \tau)}{\sum_{j=1}^C \exp(\mathbf{z}_i^{e\top} \boldsymbol{\mu}_j / \tau)}$

      `// calculate the loss for high separation`

10      $\mathcal{L}_{\text{sep}} = \frac{1}{C} \sum_{i=1}^C \log \frac{1}{C-1} \sum_{j \neq i, j \in \mathcal{Y}} \exp\left(\boldsymbol{\mu}_i^\top \boldsymbol{\mu}_j / \tau\right)$

      `// calculate overall loss`

11      $\mathcal{L} = \mathcal{L}_{\text{var}} + \mathcal{L}_{\text{sep}}$

      `// update the network weights`

12      update the weights in the deep neural network

---

## B  BROADER IMPACTS

Our work facilitates the theoretical understanding of OOD generalization through prototypical learning, which encourages low variation and high separation in the hyperspherical space. In Section 5.2, we qualitatively and quantitatively verify the low intra-class variation of the learned embeddings and we discuss in Section 6 that the variation estimate determines the general upper bound on the generalization error for a learnable OOD generalization task. This provable framework may serve as a foothold that can be useful for future OOD generalization research via representation learning.

From a practical viewpoint, our research can directly impact many real applications, when deploying machine learning models in the real world. Out-of-distribution generalization is a fundamental problem and is commonly encountered when building reliable ML systems in the industry. Our empirical results show that our approach achieves consistent improvement over the baseline on a wide range of tasks. Overall, our work has both theoretical and practical impacts.

## C  THEORETICAL ANALYSIS

**Notations.** We first set up notations for theoretical analysis. Recall that $\mathbb{P}^e_{XY}$ denotes the joint distribution of $X, Y$ in domain $e$. The label set $\mathcal{Y} := \{1, 2, \cdots, C\}$. For an input $\mathbf{x}$, $\mathbf{z} = h(\mathbf{x})/\|h(\mathbf{x})\|_2$ is its feature embedding. Let $\mathbb{P}^{e,y}_X$ denote the marginal distribution of $X$ in domain $e$ with class $y$. Similarly, $\mathbb{P}^{e,y}_Z$ denotes the marginal distribution of $Z$ in domain $e$ with class $y$. Let $E := |\mathcal{E}_{\text{train}}|$ for abbreviation. As we do not consider the existence of spurious correlation in this work, it is natural to assume that domains and classes are uniformly distributed: $\mathbb{P}_X := \frac{1}{EC} \sum_{e,y} \mathbb{P}^{e,y}_X$. We specify the

distance metric to be the Wasserstein-1 distance *i.e.,* $\mathcal{W}_1(\cdot, \cdot)$ and define all notions of variation under such distance.

Next, we proceed with several lemmas that are particularly useful to prove our main theorem.

**Lemma C.1.** *With probability at least* $1 - \delta$,

$$-\mathbb{E}_{(\mathbf{x},c)\sim\mathbb{P}_{XY}}\boldsymbol{\mu}_c^\top \frac{h(\mathbf{x})}{\|h(\mathbf{x})\|_2} + \frac{1}{N}\sum_{i=1}^N \boldsymbol{\mu}_{c(i)}^\top \frac{h(\mathbf{x}_i)}{\|h(\mathbf{x}_i)\|_2} \le \mathbb{E}_{S\sim\mathbb{P}_N}[\frac{1}{N}\mathbb{E}_{\sigma_1,\dots,\sigma_N}\sup_{h\in\mathcal{H}}\sum_{i=1}^N \sigma_i\boldsymbol{\mu}_{c(i)}^\top \frac{h(\mathbf{x}_i)}{\|h(\mathbf{x}_i)\|_2}] + \beta\sqrt{\frac{\ln(2/\delta)}{N}}.$$

*where* $\beta$ *is a universal constant and* $\sigma_1,\dots,\sigma_N$ *are Rademacher variables.*

*Proof.* By Cauchy-Schwarz inequality,

$$|\boldsymbol{\mu}_{c(i)}^\top \frac{h(\mathbf{x}_i)}{\|h(\mathbf{x}_i)\|_2}| \le \|\boldsymbol{\mu}_{c(i)}\|_2 \left\|\frac{h(\mathbf{x}_i)}{\|h(\mathbf{x}_i)\|_2}\right\|_2 = 1$$

Define $\mathcal{G} = \{\langle \frac{h(\cdot)}{\|h(\cdot)\|_2}, \cdot\rangle : h \in \mathcal{H}\}$. Let $S = (\mathbf{u}_1,\dots,\mathbf{u}_N) \sim \mathbb{P}_N$ where $\mathbf{u}_i = \begin{pmatrix}\mathbf{x}_i \\ \boldsymbol{\mu}_{c(i)}\end{pmatrix}$ and $N$ is the sample size. The Rademacher complexity of $\mathcal{G}$ is

$$\mathcal{R}_N(\mathcal{G}) := \mathbb{E}_{S\sim\mathbb{P}_N}[\frac{1}{N}\sup_{g\in\mathcal{G}}\sum_{i=1}^N \sigma_i g(\mathbf{u}_i)].$$

We can apply the standard Rademacher complexity bound (Theorem 26.5 in Shalev-Shwartz and Ben-David) to $\mathcal{G}$, then we have that,

$$-\mathbb{E}_{(\mathbf{x},c)\sim\mathbb{P}_{XY}}\boldsymbol{\mu}_c^\top \frac{h(\mathbf{x})}{\|h(\mathbf{x})\|_2} + \frac{1}{N}\sum_{i=1}^N \boldsymbol{\mu}_{c(i)}^\top \frac{h(\mathbf{x}_i)}{\|h(\mathbf{x}_i)\|_2} \le \mathbb{E}_{S\sim\mathbb{P}_N}[\frac{1}{N}\mathbb{E}_{\sigma_1,\dots,\sigma_N}\sup_{g\in\mathcal{G}}\sum_{i=1}^N \sigma_i g(\mathbf{u}_i)] + \beta\sqrt{\frac{\ln(2/\delta)}{N}}$$

$$= \mathbb{E}_{S\sim\mathbb{P}_N}[\frac{1}{N}\mathbb{E}_{\sigma_1,\dots,\sigma_N}\sup_{h\in\mathcal{H}}\sum_{i=1}^N \sigma_i\boldsymbol{\mu}_{c(i)}^\top \frac{h(\mathbf{x}_i)}{\|h(\mathbf{x}_i)\|_2}] + \beta\sqrt{\frac{\ln(2/\delta)}{N}},$$

where $\beta$ is a universal positive constant.

**Remark 1.** *The above lemma indicates that when samples are sufficiently aligned with their class prototypes on the hyperspherical feature space,* i.e., $\frac{1}{N}\sum_{i=1}^N \boldsymbol{\mu}_{c(i)}^\top \frac{h(\mathbf{x}_i)}{\|h(\mathbf{x}_i)\|_2} \ge 1 - \epsilon$ *for some small constant* $\epsilon > 0$, *we can upper bound* $-\mathbb{E}_{(\mathbf{x},c)\sim\mathbb{P}_{XY}}\boldsymbol{\mu}_c^\top \frac{h(\mathbf{x})}{\|h(\mathbf{x})\|_2}$. *This result will be useful to prove Thm 6.1.*

**Lemma C.2.** *Suppose* $\mathbb{E}_{(\mathbf{z},c)\sim\mathbb{P}_{ZY}}\boldsymbol{\mu}_c^\top\mathbf{z} \ge 1 - \gamma$. *Then, for all* $e \in \mathcal{E}_{\text{train}}$ *and* $y \in [C]$, *we have that*

$$\mathbb{E}_{\mathbf{z}\sim\mathbb{P}_Z^{e,y}}\boldsymbol{\mu}_c^\top\mathbf{z} \ge 1 - CE\gamma.$$

*Proof.* Fix $e' \in \mathcal{E}_{\text{train}}$ and $y' \in [C]$. Then,

$$1 - \gamma \le \mathbb{E}_{(\mathbf{z},c)\sim\mathbb{P}_{ZY}}\boldsymbol{\mu}_c^\top\mathbf{z}$$

$$= \frac{1}{CE}\sum_{e\in\mathcal{E}_{\text{train}}}\sum_{y\in[C]}\mathbb{E}_{\mathbf{z}\sim\mathbb{P}_Z^{e,y}}\mathbf{z}^\top\boldsymbol{\mu}_y$$

$$= \frac{1}{CE}\mathbb{E}_{\mathbf{z}\sim\mathbb{P}_Z^{e',y'}}\mathbf{z}^\top\boldsymbol{\mu}_{y'} + \frac{1}{CE}\sum_{(e,y)\in\mathcal{E}_{\text{train}}\times[C]\setminus\{(e',y')\}}\mathbb{E}_{\mathbf{z}\sim\mathbb{P}_Z^{e,y}}\mathbf{z}^\top\boldsymbol{\mu}_y$$

$$\le \frac{1}{CE}\mathbb{E}_{\mathbf{z}\sim\mathbb{P}_Z^{e',y'}}\mathbf{z}^\top\boldsymbol{\mu}_{y'} + \frac{CE - 1}{CE}$$

where the last line holds by $|\mathbf{z}^\top\boldsymbol{\mu}_c| \le 1$ and we also used the assumption that the domains and classes are uniformly distributed. Rearranging the terms, we have

$$1 - CE\gamma \le \mathbb{E}_{\mathbf{z}\sim\mathbb{P}_Z^{e',y'}}\mathbf{z}^\top\boldsymbol{\mu}_{y'}$$

**Lemma C.3.** *Fix $y \in [C]$ and $e \in \mathcal{E}_{\text{train}}$. Fix $\eta > 0$. If $\mathbb{E}_{\mathbf{z} \sim \mathbb{P}_Z^{e,y}} \mathbf{z}^\top \boldsymbol{\mu}_y \geq 1 - CE\gamma$, then*

$$\mathbb{P}_Z^{e,y}(\|\mathbf{z} - \boldsymbol{\mu}_y\|_2 \geq \eta) \leq \frac{2CE\gamma}{\eta^2}.$$

*Proof.* Note that

$$\|\mathbf{z} - \boldsymbol{\mu}_y\|_2^2 = \|\mathbf{z}\|_2^2 + \|\boldsymbol{\mu}_y\|_2^2 - 2\mathbf{z}^\top \boldsymbol{\mu}_y$$
$$= 2 - 2\mathbf{z}^\top \boldsymbol{\mu}_y.$$

Taking the expectation on both sides and applying the hypothesis, we have that

$$\mathbb{E}_{\mathbf{z} \sim \mathbb{P}_Z^{e,y}} \|\mathbf{z} - \boldsymbol{\mu}_c\|_2^2 \leq 2CE\gamma.$$

Applying Chebyschev's inequality to $\|\mathbf{z} - \boldsymbol{\mu}_y\|_2$, we have that

$$\mathbb{P}_Z^{e,y}(\|\mathbf{z} - \boldsymbol{\mu}_y\|_2 \geq \eta) \leq \frac{\text{Var}(\|\mathbf{z} - \boldsymbol{\mu}_y\|_2)}{\eta^2}$$
$$\leq \frac{\mathbb{E}_{\mathbf{z} \sim \mathbb{P}_Z^{e,y}}(\|\mathbf{z} - \boldsymbol{\mu}_y\|_2^2)}{\eta^2}$$
$$\leq \frac{2CE\gamma}{\eta^2}$$

**Lemma C.4.** *Fix $y \in [C]$. Fix $e, e' \in \mathcal{E}_{\text{train}}$. Suppose $\mathbb{E}_{\mathbf{z} \sim \mathbb{P}_Z^{e,y}} \mathbf{z}^\top \boldsymbol{\mu}_c \geq 1 - CE\gamma$. Fix $\mathbf{v} \in S^{d-1}$. Let $P$ denote the distribution of $\mathbf{v}^\top \mathbf{z}_e$ and $Q$ denote the distribution $\mathbf{v}^\top \mathbf{z}_{e'}$. Then,*

$$\mathcal{W}_1(P, Q) \leq 10(CE\gamma)^{1/3}$$

*where $\mathcal{W}_1(P, Q)$ is the Wassersisten-1 distance.*

*Proof.* Consider the dual formulation of Wasserstein-1 distance:

$$\mathcal{W}(P, Q) = \sup_{f : \|f\|_{\text{lip}} \leq 1} \mathbb{E}_{\mathbf{x} \sim \mathbb{P}_X^{e,y}}[f(\mathbf{v}^\top \mathbf{x})] - \mathbb{E}_{\mathbf{x} \sim \mathbb{P}_X^{e',y}}[f(\mathbf{v}^\top \mathbf{x})]$$

where $\|f\|_{\text{lip}}$ denotes the Lipschitz norm. Let $\kappa > 0$. There exists $f_0$ such that

$$\mathcal{W}(P, Q) \leq \mathbb{E}_{\mathbf{z} \sim \mathbb{P}_Z^{e,y}}[f_0(\mathbf{v}^\top \mathbf{z})] - \mathbb{E}_{\mathbf{z} \sim \mathbb{P}_Z^{e',y}}[f_0(\mathbf{v}^\top \mathbf{z})] + \kappa.$$

We assume that without loss of generality $f_0(\boldsymbol{\mu}_y^\top \mathbf{v}) = 0$. Define $f'(\cdot) = f_0(\cdot) - f_0(\boldsymbol{\mu}_y^\top \mathbf{v})$. Then, note that $f'(\boldsymbol{\mu}_y^\top \mathbf{v}) = 0$ and

$$\mathbb{E}_{\mathbf{z} \sim \mathbb{P}_Z^{e,y}}[f'(\mathbf{v}^\top \mathbf{z})] - \mathbb{E}_{z \sim \mathbb{P}_Z^{e',y}}[f'(\mathbf{v}^\top \mathbf{z})] = \mathbb{E}_{\mathbf{z} \sim \mathbb{P}_Z^{e,y}}[f_0(\mathbf{v}^\top \mathbf{z})] - \mathbb{E}_{\mathbf{z} \sim \mathbb{P}_Z^{e',y}}[f_0(\mathbf{v}^\top \mathbf{z})] + f'(\boldsymbol{\mu}_y^\top v) - f'(\boldsymbol{\mu}_y^\top v)$$
$$= \mathbb{E}_{\mathbf{z} \sim \mathbb{P}_Z^{e,y}}[f_0(\mathbf{v}^\top \mathbf{z})] - \mathbb{E}_{\mathbf{z} \sim \mathbb{P}_Z^{e',y}}[f_0(\mathbf{v}^\top \mathbf{z})],$$

proving the claim.

Now define $B := \{\mathbf{u} \in S^{d-1} : \|\mathbf{u} - \boldsymbol{\mu}_y\|_2 \leq \eta\}$. Then, we have

$$\mathbb{E}_{\mathbf{z} \sim \mathbb{P}_Z^{e,y}}[f_0(\mathbf{v}^\top \mathbf{z})] - \mathbb{E}_{\mathbf{z} \sim \mathbb{P}_Z^{e',y}}[f_0(\mathbf{v}^\top \mathbf{z})] = \mathbb{E}_{\mathbf{z} \sim \mathbb{P}_Z^{e,y}}[f_0(\mathbf{v}^\top \mathbf{z})\mathbb{1}\{\mathbf{z} \in B\}] - \mathbb{E}_{\mathbf{z} \sim \mathbb{P}_Z^{e',y}}[f_0(\mathbf{v}^\top \mathbf{z})\mathbb{1}\{\mathbf{z} \in B\}]$$
$$+ \mathbb{E}_{\mathbf{z} \sim \mathbb{P}_Z^{e,y}}[f_0(\mathbf{v}^\top \mathbf{z})\mathbb{1}\{\mathbf{z} \notin B\}] - \mathbb{E}_{\mathbf{z} \sim \mathbb{P}_Z^{e',y}}[f_0(\mathbf{v}^\top \mathbf{z})\mathbb{1}\{\mathbf{z} \notin B\}]$$

Note that if $\mathbf{z} \in B$, then by $\|f\|_{\text{lip}} \leq 1$,

$$|f_0(\mathbf{v}^\top \mathbf{z}) - f_0(\mathbf{v}^\top \boldsymbol{\mu}_y)| \leq |\mathbf{v}^\top(\mathbf{z} - \boldsymbol{\mu}_y)|$$
$$\leq \|\mathbf{v}\|_2 \|\mathbf{z} - \boldsymbol{\mu}_y\|_2$$
$$\leq \eta.$$

Therefore, $|f_0(\mathbf{v}^\top \mathbf{z})| \leq \eta$ and we have that

$$\mathbb{E}_{\mathbf{z} \sim \mathbb{P}_Z^{e,y}}[f_0(\mathbf{v}^\top \mathbf{z}) \mathbb{1}\{\mathbf{z} \in B\}] - \mathbb{E}_{\mathbf{z} \sim \mathbb{P}_Z^{e',y}}[f_0(\mathbf{v}^\top \mathbf{z}) \mathbb{1}\{\mathbf{z} \in B\}] \leq 2\eta(\mathbb{E}_{\mathbf{z} \sim \mathbb{P}_Z^{e,y}}[\mathbb{1}\{\mathbf{z} \in B\}] + \mathbb{E}_{\mathbf{z} \sim \mathbb{P}_Z^{e',y}}[\mathbb{1}\{\mathbf{z} \in B\}])$$
$$\leq 2\eta.$$

Now, note that $\max_{\mathbf{u} \in S^{d-1}} |f(\mathbf{u}^\top \mathbf{v})| \leq 2$ (repeat the argument from above but use $\|\mathbf{u} - \boldsymbol{\mu}_y\|_2 \leq 2$. Then,

$$\mathbb{E}_{\mathbf{z} \sim \mathbb{P}_Z^{e,y}}[f_0(\mathbf{v}^\top \mathbf{z}) \mathbb{1}\{\mathbf{z} \notin B\}] - \mathbb{E}_{\mathbf{z} \sim \mathbb{P}_Z^{e',y}}[f_0(\mathbf{v}^\top \mathbf{z}) \mathbb{1}\{\mathbf{z} \notin B\}] \leq 2[\mathbb{E}_{\mathbf{z} \sim \mathbb{P}_Z^{e,y}}[\mathbb{1}\{\mathbf{z} \notin B\}] + \mathbb{E}_{\mathbf{z} \sim \mathbb{P}_Z^{e',y}}[\mathbb{1}\{\mathbf{z} \notin B\}]]$$
$$\leq \frac{8CE\gamma}{\eta}$$

where in the last line, we used the hypothesis and Lemma C.3. Thus, by combining the above, we have that

$$\mathcal{W}(P, Q) \leq 2\eta + \frac{8CE\gamma}{\eta^2} + \kappa.$$

Choosing $\eta = (CE\gamma)^{1/3}$, we have that

$$\mathcal{W}(P, Q) \leq 10(CE\gamma)^{1/3} + \kappa.$$

Since $\kappa > 0$ was arbitrary, we can let it go to 0, obtaining the result.

Next, we are ready to prove our main results. For completeness, we state the theorem here.

**Theorem C.1** (Variation upper bound (Thm 4.1)). *Suppose samples are aligned with class prototypes such that $\frac{1}{N} \sum_{j=1}^N \boldsymbol{\mu}_{c(j)}^\top \mathbf{z}_j \geq 1 - \epsilon$ for some $\epsilon \in (0, 1)$, where $\mathbf{z}_j = \frac{h(\mathbf{x}_j)}{\|h(\mathbf{x}_j)\|_2}$. Then $\exists \delta \in (0, 1)$, with probability at least $1 - \delta$,*

$$\mathcal{V}^{\sup}(h, \Sigma_{\mathrm{avail}}) \leq O(\epsilon^{1/3} + (\mathbb{E}_{\mathcal{D}}[\frac{1}{N}\mathbb{E}_{\sigma_1,\dots,\sigma_N} \sup_{h \in \mathcal{H}} \sum_{i=1}^N \sigma_i \mathbf{z}_i^\top \boldsymbol{\mu}_{c(i)}])^{1/3} + (\frac{\ln(2/\delta)}{N})^{1/6}),$$

*where $\sigma_1, \dots, \sigma_N$ are Rademacher random variables and $O(\cdot)$ suppresses dependence on constants and $|\mathcal{E}_{\mathrm{avail}}|$.*

*Proof of Theorem 6.1.* Suppose $\frac{1}{N} \sum_{j=1}^N \boldsymbol{\mu}_{c(j)}^\top \mathbf{z}_j = \frac{1}{N} \sum_{i=1}^N \boldsymbol{\mu}_{c(i)}^\top \frac{h(\mathbf{x}_i)}{\|h(\mathbf{x}_i)\|_2} \geq 1 - \epsilon$. Then, by Lemma C.1, with probability at least $1 - \delta$, we have

$$-\mathbb{E}_{(\mathbf{x},c) \sim \mathbb{P}_{XY}} \boldsymbol{\mu}_c^\top \frac{h(\mathbf{x})}{\|h(\mathbf{x})\|_2} \leq \mathbb{E}_{S \sim \mathbb{P}_N}[\frac{1}{N}\mathbb{E}_{\sigma_1,\dots,\sigma_N} \sup_{h \in \mathcal{H}} \sum_{i=1}^N \sigma_i \boldsymbol{\mu}_{c(i)}^\top \frac{h(\mathbf{x}_i)}{\|h(\mathbf{x}_i)\|_2}] + \beta\sqrt{\frac{\ln(2/\delta)}{N}} - \frac{1}{N}\sum_{i=1}^N \boldsymbol{\mu}_{c(i)}^\top \frac{h(\mathbf{x}_i)}{\|h(\mathbf{x}_i)\|_2}$$
$$\leq \mathbb{E}_{S \sim \mathbb{P}_N}[\frac{1}{N}\mathbb{E}_{\sigma_1,\dots,\sigma_N} \sup_{h \in \mathcal{H}} \sum_{i=1}^N \sigma_i \boldsymbol{\mu}_{c(i)}^\top \frac{h(\mathbf{x}_i)}{\|h(\mathbf{x}_i)\|_2}] + \beta\sqrt{\frac{\ln(2/\delta)}{N}} + \epsilon - 1$$

where $\sigma_1, \dots, \sigma_N$ denote Rademacher random variables and $\beta$ is a universal positive constant. Define $\gamma = \epsilon + \mathbb{E}_{S \sim \mathbb{P}_N}[\frac{1}{N}\mathbb{E}_{\sigma_1,\dots,\sigma_N} \sup_{h \in \mathcal{H}} \sum_{i=1}^N \sigma_i \boldsymbol{\mu}_{c(i)}^\top \frac{h(\mathbf{x}_i)}{\|h(\mathbf{x}_i)\|_2}] + \beta\sqrt{\frac{\ln(2/\delta)}{N}}$. Then, we have

$$\mathbb{E}_{(\mathbf{z},c) \sim \mathbb{P}_{ZY}} \boldsymbol{\mu}_c^\top \mathbf{z} \geq 1 - \gamma.$$

Then, by Lemma C.2, for all $e \in \mathcal{E}_{\mathrm{train}}$ and $y \in [C]$,

$$\mathbb{E}_{\mathbf{z} \sim \mathbb{P}_Z^{e,y}} \boldsymbol{\mu}_y^\top \mathbf{z} \geq 1 - CE\gamma.$$

Let $\alpha > 0$ and $\mathbf{v}_0$ such that

$$\mathcal{V}^{\sup}(h, \mathcal{E}_{\mathrm{train}}) = \sup_{\mathbf{v} \in S^{d-1}} \mathcal{V}(\mathbf{v}^\top h, \mathcal{E}_{\mathrm{train}}) \leq \mathcal{V}(\mathbf{v}_0^\top h, \mathcal{E}_{\mathrm{train}}) + \alpha$$

Let $Q_{\mathbf{v}_0}^{e,y}$ denote the distribution of $\mathbf{v}_0^\top \mathbf{z}$ in domain $e$ under class $y$. From Lemma C.4, we have that

$$\mathcal{W}_1(Q_{\mathbf{v}_0}^{e,y}, Q_{\mathbf{v}_0}^{',y}) \leq 10(CE\gamma)^{1/3}$$

for all $y \in [C]$ and $e, e' \in \mathcal{E}_{\text{train}}$.

We have that

$$
\begin{aligned}
\sup_{\mathbf{v} \in S^{d-1}} \mathcal{V}(\mathbf{v}^\top h, \mathcal{E}_{\text{train}}) &= \sup_{\mathbf{v} \in S^{d-1}} \mathcal{V}(\mathbf{v}^\top h, \mathcal{E}_{\text{train}}) \\
&= \max_y \sup_{e,e'} \mathcal{W}_1(Q_{\mathbf{v}_0}^{e,y}, Q_{\mathbf{v}_0}^{e',y}) + \alpha \\
&\leq 10(CE\gamma)^{1/3} + \alpha.
\end{aligned}
$$

Noting that $\alpha$ was arbitrary, we may send it to 0 yielding

$$\sup_{\mathbf{v} \in S^{d-1}} \mathcal{V}(\mathbf{v}^\top h, \mathcal{E}_{\text{train}}) \leq 10(CE\gamma)^{1/3}.$$

Now, using the inequality that for $a, b, c \geq 0$, $(a + b + c)^{1/3} \leq a^{1/3} + b^{1/3} + c^{1/3}$, we have that

$$\mathcal{V}^{\text{sup}}(h, \mathcal{E}_{\text{train}}) \leq O(\epsilon^{1/3} + (\mathbb{E}_{S \sim \mathbb{P}_N}[\frac{1}{N}\mathbb{E}_{\sigma_1,\ldots,\sigma_N} \sup_{h \in \mathcal{H}} \sum_{i=1}^N \sigma_i \boldsymbol{\mu}_{c(i)}^\top \frac{h(\mathbf{x}_i)}{\|h(\mathbf{x}_i)\|_2}])^{1/3} + \beta(\frac{\ln(2/\delta)}{N})^{1/6})$$

**Remark 2.** *As our loss promotes alignment of sample embeddings with their class prototypes on the hyperspherical space, the above Theorem implies that when such alignment holds, we can upper bound the intra-class variation with three main factors: the optimization error $\epsilon$, the Rademacher complexity of the given neural network, and the estimation error $(\frac{\ln(2/\delta)}{N})^{1/6}$.*

## C.1 EXTENSION: FROM LOW VARIATION TO LOW OOD GENERALIZATION ERROR

Ye et al. (2021) provide OOD generalization error bounds based on the notation of variation. Therefore, bounding intra-class variation is critical to bound OOD generalization error. For completeness, we reinstate the main results in Ye et al. (2021) below, which provide both OOD generalization error upper and lower bounds based on the variation w.r.t. the training domains. Interested readers shall refer to Ye et al. (2021) for more details and illustrations.

**Definition C.1** (Expansion Function (Ye et al., 2021)). *We say a function $s : \mathbb{R}^+ \cup \{0\} \to \mathbb{R}^+ \cup \{0, +\infty\}$ is an expansion function, iff the following properties hold: 1) $s(\cdot)$ is monotonically increasing and $s(x) \geq x, \forall x \geq 0$; 2) $\lim_{x \to 0^+} s(x) = s(0) = 0$.*

As it is impossible to generalize to an arbitrary distribution, characterizing the relation between $\mathcal{E}_{\text{avail}}$ and $\mathcal{E}_{\text{all}}$ is essential to formalize OOD generalization. Based on the notion of expansion function, the learnability of OOD generalization is defined as follows:

**Definition C.2** (OOD-Learnability (Ye et al., 2021)). *Let $\Phi$ be the feature space and $\rho$ be a distance metric on distributions. We say an OOD generalization problem from $\mathcal{E}_{\text{avail}}$ to $\mathcal{E}_{\text{all}}$ is* learnable *if there exists an expansion function $s(\cdot)$ and $\delta \geq 0$, such that: for all $\phi \in \Phi^2$ satisfying $\mathcal{I}_\rho(\phi, \mathcal{E}_{\text{avail}}) \geq \delta$, we have $s(\mathcal{V}_\rho(\phi, \mathcal{E}_{\text{avail}})) \geq \mathcal{V}_\rho(\phi, \mathcal{E}_{\text{all}})$. If such $s(\cdot)$ and $\delta$ exist, we further call this problem $(s(\cdot), \delta)$-learnable.*

For learnable OOD generalization problems, the following two theorems characterize OOD error upper and lower bounds based on variation.

**Theorem C.2** (OOD Error Upper Bound (Ye et al., 2021)). *Suppose we have learned a classifier with loss function $\ell(\cdot, \cdot)$ such that $\forall e \in \mathcal{E}_{\text{all}}$ and $\forall y \in \mathcal{Y}$, $p_{h^e|Y^e}(h|y) \in L^2(\mathbb{R}^d)$. $h(\cdot) \in \mathbb{R}^d$ denotes the feature extractor. Denote the characteristic function of random variable $h^e|Y^e$ as*

---

[2] $\phi$ referred to as feature $h$ in theoretical analysis.

$\hat{p}_{h^e|Y^e}(t|y) = \mathbb{E}[\exp\{i\langle t, h^e\rangle\}|Y^e = y]$. *Assume the hypothetical space $\mathcal{F}$ satisfies the following regularity conditions that $\exists \alpha, M_1, M_2 > 0, \forall f \in \mathcal{F}, \forall e \in \mathcal{E}_{\text{all}}, y \in \mathcal{Y}$,*

$$\int_{h \in \mathbb{R}^d} p_{h^e|Y^e}(h|y)|h|^\alpha \mathrm{d}h \leq M_1 \quad \text{and} \quad \int_{t \in \mathbb{R}^d} |\hat{p}_{h^e|Y^e}(t|y)||t|^\alpha \mathrm{d}t \leq M_2. \tag{8}$$

*If $(\mathcal{E}_{\text{avail}}, \mathcal{E}_{\text{all}})$ is $\big(s(\cdot), \mathcal{I}^{\inf}(h, \mathcal{E}_{\text{avail}})\big)$-learnable under $\Phi$ with Total Variation $\rho$[3], then we have*

$$\text{err}(f) \leq O\Big(s\big(\mathcal{V}^{\sup}(h, \mathcal{E}_{\text{avail}})\big)^{\frac{\alpha^2}{(\alpha+d)^2}}\Big), \tag{9}$$

*where $O(\cdot)$ depends on $d, C, \alpha, M_1, M_2$.*

**Theorem C.3** (OOD Error Lower Bound (Ye et al., 2021)). *Consider 0-1 loss: $\ell(\hat{y}, y) = \mathbb{I}(\hat{y} \neq y)$. For any $\delta > 0$ and any expansion function satisfying 1) $s'_+(0) \triangleq \lim_{x \to 0^+} \frac{s(x)-s(0)}{x} \in (1, +\infty)$; 2) exists $k > 1, t > 0$, s.t. $kx \leq s(x) < +\infty, x \in [0, t]$, there exists a constant $C_0$ and an OOD generalization problem $(\mathcal{E}_{\text{avail}}, \mathcal{E}_{\text{all}})$ that is $(s(\cdot), \delta)$-learnable under linear feature space $\Phi$ w.r.t symmetric KL-divergence $\rho$, s.t. $\forall \varepsilon \in [0, \frac{t}{2}]$, the optimal classifier $f$ satisfying $\mathcal{V}^{\sup}(h, \mathcal{E}_{\text{avail}}) = \varepsilon$ will have the OOD generalization error lower bounded by*

$$\text{err}(f) \geq C_0 \cdot s(\mathcal{V}^{\sup}(h, \mathcal{E}_{\text{avail}})). \tag{10}$$

## D   ADDITIONAL EXPERIMENTAL DETAILS

**Software and hardware.**   Our method is implemented with PyTorch 1.10. All experiments are conducted on NVIDIA GeForce RTX 2080 Ti GPUs for small to medium batch sizes and NVIDIA A100 and RTX A6000 GPUs for large batch sizes.

**Architecture.**   In our experiments, we use ResNet-18 for CIFAR-10, ResNet-34 for ImageNet-100, ResNet-50 for PACS, VLCS, Office-Home and Terra Incognita. Following common practice in prior works (Khosla et al., 2020), we use a non-linear MLP projection head to obtain features in our experiments. The embedding dimension is 128 of the projection head for ImageNet-100. The projection head dimension is 512 for PACS, VLCS, Office-Home, and Terra Incognita.

**Additional implementation details.**   In our experiments, we follow the common practice that initializing the network with ImageNet pre-trained weights for PACS, VLCS, Office-Home, and Terra Incognita. We then fine-tune the network for 50 epochs. For the large-scale experiments on ImageNet-100, we fine-tune ImageNet pre-trained ResNet-34 with our method for 10 epochs for computational efficiency. We set the temperature $\tau = 0.1$, prototype update factor $\alpha = 0.95$ as the default value. We use stochastic gradient descent with momentum $0.9$, and weight decay $10^{-4}$. The search distribution in our experiments for the learning rate hyperparameter is: $\text{lr} \in \{0.005, 0.002, 0.001, 0.0005, 0.0002, 0.0001, 0.00005\}$. The search space for the batch size is $\text{bs} \in \{32, 64\}$. The loss weight $\lambda$ for balancing our loss function ($\mathcal{L} = \lambda\mathcal{L}_{\text{var}} + \mathcal{L}_{\text{sep}}$) is selected from $\lambda \in \{1.0, 2.0, 4.0\}$. For multi-source domain generalization, hard negatives can be incorporated by a simple modification to the denominator of the variation loss:

$$\mathcal{L}_{\text{var}} = -\frac{1}{N} \sum_{e \in \mathcal{E}_{\text{avail}}} \sum_{i=1}^{|\mathcal{D}^e|} \log \frac{\exp\big(\mathbf{z}_i^\top \boldsymbol{\mu}_{c(i)}/\tau\big)}{\sum_{j=1}^C \exp\big(\mathbf{z}_i^\top \boldsymbol{\mu}_j/\tau\big) + \sum_{j=1}^N \mathbb{I}(y_j \neq y_i, e_i = e_j) \exp\big(\mathbf{z}_i^\top \mathbf{z}_j/\tau\big)} \tag{11}$$

**Details of datasets.**   We provide a detailed description of the datasets used in this work:

**CIFAR-10** (Krizhevsky et al., 2009) is consist of $60,000$ color images with 10 classes. The training set has $50,000$ images and the test set has $10,000$ images.

**ImageNet-100** is composed by randomly sampled 100 categories from ImageNet-1K. This dataset contains the following classes: n01498041, n01514859, n01582220, n01608432, n01616318, n01687978, n01776313, n01806567, n01833805, n01882714, n01910747, n01944390, n01985128, n02007558, n02071294,

---

[3]For two distribution $\mathbb{P}, \mathbb{Q}$ with probability density function $p, q$, $\rho(\mathbb{P}, \mathbb{Q}) = \frac{1}{2}\int_x |p(x) - q(x)|\mathrm{d}x$.

n02085620, n02114855, n02123045, n02128385, n02129165, n02129604, n02165456, n02190166, n02219486, n02226429, n02279972, n02317335, n02326432, n02342885, n02363005, n02391049, n02395406, n02403003, n02422699, n02442845, n02444819, n02480855, n02510455, n02640242, n02672831, n02687172, n02701002, n02730930, n02769748, n02782093, n02787622, n02793495, n02799071, n02802426, n02814860, n02840245, n02906734, n02948072, n02980441, n02999410, n03014705, n03028079, n03032252, n03125729, n03160309, n03179701, n03220513, n03249569, n03291819, n03384352, n03388043, n03450230, n03481172, n03594734, n03594945, n03627232, n03642806, n03649909, n03661043, n03676483, n03724870, n03733281, n03759954, n03761084, n03773504, n03804744, n03916031, n03938244, n04004767, n04026417, n04090263, n04133789, n04153751, n04296562, n04330267, n04371774, n04404412, n04465501, n04485082, n04507155, n04536866, n04579432, n04606251, n07714990, n07745940.

**CIFAR-10-C** is generated based on the previous literature (Hendrycks & Dietterich, 2019), applying different corruptions on CIFAR-10 data. The corruption types include gaussian noise, zoom blur, impulse noise, defocus blur, snow, brightness, contrast, elastic transform, fog, frost, gaussian blur, glass blur, JEPG compression, motion blur, pixelate, saturate, shot noise, spatter, and speckle noise.

**ImageNet-100-C** is algorithmically generated with Gaussian noise based on (Hendrycks & Dietterich, 2019) for the ImageNet-100 dataset.

**PACS** (Li et al., 2017) is commonly used in OoD generalization. This dataset contains $9,991$ examples of resolution $224 \times 224$ and four domains with different image styles, namely photo, art painting, cartoon, and sketch with seven categories.

**VLCS** (Gulrajani & Lopez-Paz, 2020) comprises four domains including Caltech101, LabelMe, SUN09, and VOC2007. It contains $10,729$ examples of resolution $224 \times 224$ and 5 classes.

**Office-Home** (Gulrajani & Lopez-Paz, 2020) contains four different domains: art, clipart, product, and real. This dataset comprises $15,588$ examples of resolution $224 \times 224$ and 65 classes.

**Terra Incognita** (Gulrajani & Lopez-Paz, 2020) comprises images of wild animals taken by cameras at four different locations: location100, location38, location43, and location46. This dataset contains $24,788$ examples of resolution $224 \times 224$ and 10 classes.

# E DETAILED RESULTS ON CIFAR-10

In this section, we provide complete results of the different corruption types on CIFAR-10. In Table 3, we evaluate HYPO under various common corruptions. Results suggest that HYPO achieves consistent improvement over the ERM baseline for all 19 different corruptions. We also compare our loss (HYPO) with more recent competitive algorithms: EQRM (Eastwood et al., 2022) and SharpDRO (Huang et al., 2023), on the CIFAR10-C dataset (Gaussian noise). The results on ResNet-18 are presented in Table 15.

Table 3: Main results for verifying OOD generalization performance on the 19 different covariate shifts datasets. We train on CIFAR-10 as ID, using CIFAR-10-C as the OOD test dataset. Acc. denotes the accuracy on the OOD test set.

| Method | Corruptions | Acc. | Corruptions | Acc. | Corruptions | Acc. | Corruptions | Acc. |
|---|---|---|---|---|---|---|---|---|
| **CE** | Gaussian noise | 78.09 | Zoom blur | 88.47 | Impulse noise | 83.60 | Defocus blur | 94.85 |
| **HYPO (Ours)** | Gaussian noise | 85.21 | Zoom blur | 93.28 | Impulse noise | 87.54 | Defocus blur | 94.90 |
| **CE** | Snow | 90.19 | Brightness | 94.83 | Contrast | 94.11 | Elastic transform | 90.36 |
| **HYPO (Ours)** | Snow | 91.10 | Brightness | 94.87 | Contrast | 94.53 | Elastic transform | 91.64 |
| **CE** | Fog | 94.45 | Frost | 90.33 | Gaussian blur | 94.85 | Glass blur | 56.99 |
| **HYPO (Ours)** | Fog | 94.57 | Frost | 92.28 | Gaussian blur | 94.91 | Glass blur | 63.66 |
| **CE** | JEPG compression | 86.95 | Motion blur | 90.69 | Pixelate | 92.67 | Saturate | 92.86 |
| **HYPO (Ours)** | JEPG compression | 89.24 | Motion blur | 93.07 | Pixelate | 93.95 | Saturate | 93.66 |
| **CE** | Shot noise | 85.86 | Spatter | 92.20 | Speckle noise | 85.66 | **Average** | 88.32 |
| **HYPO (Ours)** | Shot noise | 89.87 | Spatter | 92.46 | Speckle noise | 89.94 | **Average** | **90.56** |

| Algorithm | Art painting | Cartoon | Photo | Sketch | Average Acc. (%) |
|---|---|---|---|---|---|
| **IRM** (Arjovsky et al., 2019) | 84.8 | 76.4 | 96.7 | 76.1 | 83.5 |
| **DANN** (Ganin et al., 2016) | 86.4 | 77.4 | 97.3 | 73.5 | 83.7 |
| **CDANN** (Li et al., 2018c) | 84.6 | 75.5 | 96.8 | 73.5 | 82.6 |
| **GroupDRO** (Sagawa et al., 2020) | 83.5 | 79.1 | 96.7 | 78.3 | 84.4 |
| **MTL** (Blanchard et al., 2021) | 87.5 | 77.1 | 96.4 | 77.3 | 84.6 |
| **I-Mixup** (Wang et al., 2020; Xu et al., 2020; Yan et al., 2020) | 86.1 | 78.9 | 97.6 | 75.8 | 84.6 |
| **MMD** (Li et al., 2018b) | 86.1 | 79.4 | 96.6 | 76.5 | 84.7 |
| **VREx** (Krueger et al., 2021) | 86.0 | 79.1 | 96.9 | 77.7 | 84.9 |
| **MLDG** (Li et al., 2018a) | 85.5 | 80.1 | 97.4 | 76.6 | 84.9 |
| **ARM** (Zhang et al., 2021) | 86.8 | 76.8 | 97.4 | 79.3 | 85.1 |
| **RSC** (Huang et al., 2020) | 85.4 | 79.7 | 97.6 | 78.2 | 85.2 |
| **Mixstyle** (Zhou et al., 2021) | 86.8 | 79.0 | 96.6 | 78.5 | 85.2 |
| **ERM** (Vapnik, 1999) | 84.7 | 80.8 | 97.2 | 79.3 | 85.5 |
| **CORAL** (Sun & Saenko, 2016) | 88.3 | 80.0 | 97.5 | 78.8 | 86.2 |
| **SagNet** (Nam et al., 2021) | 87.4 | 80.7 | 97.1 | 80.0 | 86.3 |
| **SelfReg** (Kim et al., 2021) | 87.9 | 79.4 | 96.8 | 78.3 | 85.6 |
| **GVRT** Min et al. (2022) | 87.9 | 78.4 | 98.2 | 75.7 | 85.1 |
| **VNE** (Kim et al., 2023) | 88.6 | 79.9 | 96.7 | 82.3 | 86.9 |
| **HYPO (Ours)** | 87.2 | 82.3 | 98.0 | 84.5 | **88.0** |

Table 4: Comparison with state-of-the-art methods on the PACS benchmark. All methods are trained on ResNet-50. The model selection is based on a training domain validation set. To isolate the effect of loss functions, all methods are optimized using standard SGD. *Results based on retraining of PCL with SGD using official implementation. PCL with SWAD optimization is further compared in Table 2. We run HYPO 3 times and report the average and std. $\pm x$ denotes the standard error, rounded to the first decimal point.

| Algorithm | Art | Clipart | Product | Real World | Average Acc. (%) |
|---|---|---|---|---|---|
| **IRM** (Arjovsky et al., 2019) | 58.9 | 52.2 | 72.1 | 74.0 | 64.3 |
| **DANN** (Ganin et al., 2016) | 59.9 | 53.0 | 73.6 | 76.9 | 65.9 |
| **CDANN** (Li et al., 2018c) | 61.5 | 50.4 | 74.4 | 76.6 | 65.7 |
| **GroupDRO** (Sagawa et al., 2020) | 60.4 | 52.7 | 75.0 | 76.0 | 66.0 |
| **MTL** (Blanchard et al., 2021) | 61.5 | 52.4 | 74.9 | 76.8 | 66.4 |
| **I-Mixup** (Wang et al., 2020; Xu et al., 2020; Yan et al., 2020) | 62.4 | 54.8 | 76.9 | 78.3 | 68.1 |
| **MMD** (Li et al., 2018b) | 60.4 | 53.3 | 74.3 | 77.4 | 66.4 |
| **VREx** (Krueger et al., 2021) | 60.7 | 53.0 | 75.3 | 76.6 | 66.4 |
| **MLDG** (Li et al., 2018a) | 61.5 | 53.2 | 75.0 | 77.5 | 66.8 |
| **ARM** (Zhang et al., 2021) | 58.9 | 51.0 | 74.1 | 75.2 | 64.8 |
| **RSC** (Huang et al., 2020) | 60.7 | 51.4 | 74.8 | 75.1 | 65.5 |
| **Mixstyle** (Zhou et al., 2021) | 51.1 | 53.2 | 68.2 | 69.2 | 60.4 |
| **ERM** (Vapnik, 1999) | 63.1 | 51.9 | 77.2 | 78.1 | 67.6 |
| **CORAL** (Sun & Saenko, 2016) | 65.3 | 54.4 | 76.5 | 78.4 | 68.7 |
| **SagNet** (Nam et al., 2021) | 63.4 | 54.8 | 75.8 | 78.3 | 68.1 |
| **SelfReg** (Kim et al., 2021) | 63.6 | 53.1 | 76.9 | 78.1 | 67.9 |
| **GVRT** Min et al. (2022) | 66.3 | 55.8 | 78.2 | 80.4 | 70.1 |
| **VNE** (Kim et al., 2023) | 60.4 | 54.7 | 73.7 | 74.7 | 65.9 |
| **HYPO (Ours)** | 68.3 | 57.9 | 79.0 | 81.4 | **71.7** |

Table 5: Comparison with state-of-the-art methods on the Office-Home benchmark. All methods are trained on ResNet-50. The model selection is based on a training domain validation set. To isolate the effect of loss functions, all methods are optimized using standard SGD.

## F   ADDITIONAL EVALUATIONS ON OTHER OOD GENERALIZATION TASKS

In this section, we provide detailed results on more OOD generalization benchmarks, including Office-Home (Table 5), VLCS (Table 6), and Terra Incognita (Table 7). We observe that our approach achieves strong performance on these benchmarks. We compare our method with a collection of OOD generalization baselines such as IRM (Arjovsky et al., 2019), DANN (Ganin et al., 2016), CDANN (Li et al., 2018c), GroupDRO (Sagawa et al., 2020), MTL (Blanchard et al., 2021), I-Mixup (Zhang et al., 2018), MMD (Li et al., 2018b), VREx (Krueger et al., 2021), MLDG (Li et al., 2018a), ARM (Zhang et al., 2021), RSC (Huang et al., 2020), Mixstyle (Zhou et al., 2021), ERM (Vapnik, 1999), CORAL (Sun & Saenko, 2016), SagNet (Nam et al., 2021), SelfReg (Kim et al., 2021), GVRT Min et al. (2022), VNE (Kim et al., 2023). These methods are all loss-based and optimized using standard SGD. On the Office-Home, our method achieves an improved OOD generalization performance of **1.6**% compared to a competitive baseline (Sun & Saenko, 2016).

We also conduct experiments coupling with SWAD and achieve superior performance on OOD generalization. As shown in Table 8, Table 9, Table 10, our method consistently establish superior results

| Algorithm | Caltech101 | LabelMe | SUN09 | VOC2007 | Average Acc. (%) |
|---|---|---|---|---|---|
| **IRM** (Arjovsky et al., 2019) | 98.6 | 64.9 | 73.4 | 77.3 | 78.6 |
| **DANN** (Ganin et al., 2016) | 99.0 | 65.1 | 73.1 | 77.2 | 78.6 |
| **CDANN** (Li et al., 2018c) | 97.1 | 65.1 | 70.7 | 77.1 | 77.5 |
| **GroupDRO** (Sagawa et al., 2020) | 97.3 | 63.4 | 69.5 | 76.7 | 76.7 |
| **MTL** (Blanchard et al., 2021) | 97.8 | 64.3 | 71.5 | 75.3 | 77.2 |
| **I-Mixup** (Wang et al., 2020; Xu et al., 2020; Yan et al., 2020) | 98.3 | 64.8 | 72.1 | 74.3 | 77.4 |
| **MMD** (Li et al., 2018b) | 97.7 | 64.0 | 72.8 | 75.3 | 77.5 |
| **VREx** (Krueger et al., 2021) | 98.4 | 64.4 | 74.1 | 76.2 | 78.3 |
| **MLDG** (Li et al., 2018a) | 97.4 | 65.2 | 71.0 | 75.3 | 77.2 |
| **ARM** (Zhang et al., 2021) | 98.7 | 63.6 | 71.3 | 76.7 | 77.6 |
| **RSC** (Huang et al., 2020) | 97.9 | 62.5 | 72.3 | 75.6 | 77.1 |
| **Mixstyle** (Zhou et al., 2021) | 98.6 | 64.5 | 72.6 | 75.7 | 77.9 |
| **ERM** (Vapnik, 1999) | 97.7 | 64.3 | 73.4 | 74.6 | 77.5 |
| **CORAL** (Sun & Saenko, 2016) | 98.3 | 66.1 | 73.4 | 77.5 | 78.8 |
| **SagNet** (Nam et al., 2021) | 97.9 | 64.5 | 71.4 | 77.5 | 77.8 |
| **SelfReg** (Kim et al., 2021) | 96.7 | 65.2 | 73.1 | 76.2 | 77.8 |
| **GVRT** Min et al. (2022) | 98.8 | 64.0 | 75.2 | 77.9 | 79.0 |
| **VNE** (Kim et al., 2023) | 97.5 | 65.9 | 70.4 | 78.4 | 78.1 |
| **HYPO (Ours)** | 98.1 | 65.3 | 73.1 | 76.3 | 78.2 |

Table 6: Comparison with state-of-the-art methods on the VLCS benchmark. All methods are trained on ResNet-50. The model selection is based on a training domain validation set. To isolate the effect of loss functions, all methods are optimized using standard SGD.

| Algorithm | Location100 | Location38 | Location43 | Location46 | Average Acc. (%) |
|---|---|---|---|---|---|
| **IRM** (Arjovsky et al., 2019) | 54.6 | 39.8 | 56.2 | 39.6 | 47.6 |
| **DANN** (Ganin et al., 2016) | 51.1 | 40.6 | 57.4 | 37.7 | 46.7 |
| **CDANN** (Li et al., 2018c) | 47.0 | 41.3 | 54.9 | 39.8 | 45.8 |
| **GroupDRO** (Sagawa et al., 2020) | 41.2 | 38.6 | 56.7 | 36.4 | 43.2 |
| **MTL** (Blanchard et al., 2021) | 49.3 | 39.6 | 55.6 | 37.8 | 45.6 |
| **I-Mixup** (Wang et al., 2020; Xu et al., 2020; Yan et al., 2020) | 59.6 | 42.2 | 55.9 | 33.9 | 47.9 |
| **MMD** (Li et al., 2018b) | 41.9 | 34.8 | 57.0 | 35.2 | 42.2 |
| **VREx** (Krueger et al., 2021) | 48.2 | 41.7 | 56.8 | 38.7 | 46.4 |
| **MLDG** (Li et al., 2018a) | 54.2 | 44.3 | 55.6 | 36.9 | 47.8 |
| **ARM** (Zhang et al., 2021) | 49.3 | 38.3 | 55.8 | 38.7 | 45.5 |
| **RSC** (Huang et al., 2020) | 50.2 | 39.2 | 56.3 | 40.8 | 46.6 |
| **Mixstyle** (Zhou et al., 2021) | 54.3 | 34.1 | 55.9 | 31.7 | 44.0 |
| **ERM** (Vapnik, 1999) | 49.8 | 42.1 | 56.9 | 35.7 | 46.1 |
| **CORAL** (Sun & Saenko, 2016) | 51.6 | 42.2 | 57.0 | 39.8 | 47.7 |
| **SagNet** (Nam et al., 2021) | 53.0 | 43.0 | 57.9 | 40.4 | 48.6 |
| **SelfReg** (Kim et al., 2021) | 48.8 | 41.3 | 57.3 | 40.6 | 47.0 |
| **GVRT** Min et al. (2022) | 53.9 | 41.8 | 58.2 | 38.0 | 48.0 |
| **VNE** (Kim et al., 2023) | 58.1 | 42.9 | 58.1 | 43.5 | 50.6 |
| **HYPO (Ours)** | 58.8 | 46.6 | 58.7 | 42.7 | **51.7** |

Table 7: Comparison with state-of-the-art methods on the Terra Incognita benchmark. All methods are trained on ResNet-50. The model selection is based on a training domain validation set. To isolate the effect of loss functions, all methods are optimized using standard SGD.

| Algorithm | Art | Clipart | Product | Real World | Average Acc. (%) |
|---|---|---|---|---|---|
| **SWAD** (Cha et al., 2021) | 66.1 | 57.7 | 78.4 | 80.2 | 70.6 |
| **PCL+SWAD** (Yao et al., 2022) | 67.3 | 59.9 | 78.7 | 80.7 | 71.6 |
| **VNE+SWAD** (Kim et al., 2023) | 66.6 | 58.6 | 78.9 | 80.5 | 71.1 |
| **HYPO+SWAD (Ours)** | 68.4 | 61.3 | 81.8 | 82.4 | **73.5** |

Table 8: Results with SWAD-based optimization on the Office-Home benchmark.

| Algorithm | Caltech101 | LabelMe | SUN09 | VOC2007 | Average Acc. (%) |
|---|---|---|---|---|---|
| **SWAD** (Cha et al., 2021) | 98.8 | 63.3 | 75.3 | 79.2 | 79.1 |
| **PCL+SWAD** (Yao et al., 2022) | 95.8 | 65.4 | 74.3 | 76.2 | 77.9 |
| **VNE+SWAD** (Kim et al., 2023) | 99.2 | 63.7 | 74.4 | 81.6 | 79.7 |
| **HYPO+SWAD (Ours)** | 98.9 | 67.8 | 74.3 | 77.7 | **79.7** |

Table 9: Rresults with SWAD-based optimization on the VLCS benchmark.

on different benchmarks including VLCS, Office-Home, Terra Incognita, showing the effectiveness of our method via hyperspherical learning.

| Algorithm | Location100 | Location38 | Location43 | Location46 | Average Acc. (%) |
|---|---|---|---|---|---|
| **SWAD** (Cha et al., 2021) | 55.4 | 44.9 | 59.7 | 39.9 | 50.0 |
| **PCL+SWAD** (Yao et al., 2022) | 58.7 | 46.3 | 60.0 | 43.6 | 52.1 |
| **VNE+SWAD** (Kim et al., 2023) | 59.9 | 45.5 | 59.6 | 41.9 | 51.7 |
| **HYPO+SWAD (Ours)** | 56.8 | 61.3 | 54.0 | 53.2 | **56.3** |

Table 10: Results with SWAD-based optimization on the Terra Incognita benchmark.

# G    EXPERIMENTS ON IMAGENET-100 AND IMAGENET-100-C

In this section, we provide additional large-scale results on the ImageNet benchmark. We use ImageNet-100 as the in-distribution data and use ImageNet-100-C with Gaussian noise as OOD data in the experiments. In Figure 6, we observe our method improves OOD accuracy compared to the ERM baseline.

# H    ABLATION OF DIFFERENT LOSS TERMS

**Ablations on separation loss.**    In Table 11, we demonstrate the effectiveness of the first loss term (variation) empirically. We compare the OOD performance of our method (with separation loss) vs. our method (without separation loss). We observe our method without separation loss term can still achieve strong OOD accuracy–average 87.2% on the PACS dataset. This ablation study indicates the first term (variation) of our method plays a more important role in practice, which aligns with our theoretical analysis in Section 6 and Appendix C.

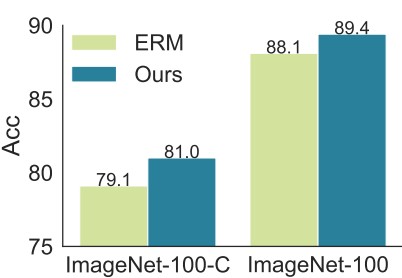

Figure 6: Experiments on ImageNet-100 (ID) vs. ImageNet-100-C (OOD).

| Algorithm | Art painting | Cartoon | Photo | Sketch | Average Acc. (%) |
|---|---|---|---|---|---|
| **Ours (w/o separation loss)** | 86.2 | 81.2 | 97.8 | 83.6 | 87.2 |
| **Ours (w separation loss)** | 87.2 | 82.3 | 98.0 | 84.5 | **88.0** |

Table 11: Ablations on separation loss term.

**Ablations on hard negative pairs.**    To verify that hard negative pairs help multiple training domains, we conduct ablation by comparing ours (with hard negative pairs) vs. ours (without hard negative pairs). We can see in Table 12 that our method with hard negative pairs improves the average OOD performance by 0.4% on the PACS dataset. Therefore, we empirically demonstrate that emphasizing hard negative pairs leads to better performance for multi-source domain generalization tasks.

| Algorithm | Art painting | Cartoon | Photo | Sketch | Average Acc. (%) |
|---|---|---|---|---|---|
| **Ours (w/o hard negative pairs)** | 87.8 | 82.9 | 98.2 | 81.4 | 87.6 |
| **Ours (w hard negative pairs)** | 87.2 | 82.3 | 98.0 | 84.5 | **88.0** |

Table 12: Ablation on hard negative pairs. OOD generalization performance on the PACS dataset.

**Comparing EMA update and learnable prototype.**    We conduct an ablation study on the prototype update rule. Specifically, we compare our method with exponential-moving-average (EMA) (Li et al., 2020; Wang et al., 2022a; Ming et al., 2023) prototype update versus learnable prototypes (LP). The results on PACS are summarized in Table 13. We observe our method with EMA achieves better average OOD accuracy 88.0% compared to learnable prototype update rules 86.7%. We empirically verify EMA-style method is a suitable prototype updating rule to facilitate gradient-based prototype update in practice.

**Quantitative verification of the $\epsilon$ factor in Theorem 6.1.**    We calculate the average intra-class variation over data from all environments $\frac{1}{N} \sum_{j=1}^{N} \boldsymbol{\mu}_{c(j)}^{\top} \mathbf{z}_j$ (Theorem 6.1) models trained with

| Algorithm | Art painting | Cartoon | Photo | Sketch | Average Acc. (%) |
|-----------|--------------|---------|-------|--------|------------------|
| **Ours (LP)** | 88.0 | 80.7 | 97.5 | 80.7 | 86.7 |
| **Ours (EMA)** | 87.2 | 82.3 | 98.0 | 84.5 | **88.0** |

Table 13: Ablation on prototype update rules. Comparing EMA update and learnable prototype (LP) on the PACS benchmark.

HYPO. Then we obtain $\hat{\epsilon} := 1 - \frac{1}{N} \sum_{j=1}^{N} \boldsymbol{\mu}_{c(j)}^{\top} \mathbf{z}_j$. We evaluated PACS, VLCS, and OfficeHome and summarized the results in Table 14. We observe that training with HYPO significantly reduces the average intra-class variation, resulting in a small epsilon ($\hat{\epsilon} < 0.1$) in practice. This suggests that the first term $O(\epsilon^{\frac{1}{3}})$ in Theorem 6.1 is indeed small for models trained with HYPO.

| Dataset | $\hat{\epsilon}$ |
|---------|------------------|
| **PACS** | 0.06 |
| **VLCS** | 0.08 |
| **OfficeHome** | 0.09 |

Table 14: Empirical verification of intra-class variation in Theorem 6.1.

| Method | OOD Acc. (%) |
|--------|--------------|
| **EQRM** (Eastwood et al., 2022) | 77.06 |
| **SharpDRO** (Huang et al., 2023) | 81.61 |
| **HYPO (ours)** | 85.21 |

Table 15: Comparison with more recent competitive baselines. Models are trained on CIFAR-10 using ResNet-18 and tested on CIFAR10-C (Gaussian noise).

# I   ANALYZING THE EFFECT OF $\tau$ AND $\alpha$

In Figure 7a, we present the OOD generalization performance by adjusting the prototype update factor $\alpha$. The results are averaged over four domains on the PACS dataset. We observe the generalization performance is competitive across a wide range of $\alpha$. In particular, our method achieves the best performance when $\alpha = 0.95$ on the PACS dataset with an average of $88.0\%$ OOD accuracy.

We show in Figure 7b the OOD generalization performance by varying the temperature parameter $\tau$. The results are averaged over four different domains on PACS. We observe a relative smaller $\tau$ results in stronger OOD performance while too large $\tau$ (e.g., 0.9) would lead to degraded performance.

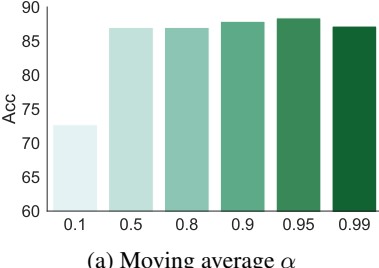
(a) Moving average $\alpha$

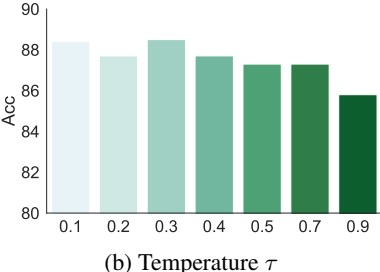
(b) Temperature $\tau$

Figure 7: Ablation on (a) prototype update discount factor $\alpha$ and (b) temperature $\tau$. The results are averaged over four domains on the PACS dataset.

## J  THEORETICAL INSIGHTS ON INTER-CLASS SEPARATION

To gain theoretical insights into inter-class separation, we focus on the learned prototype embeddings of the separation loss with a simplified setting where we directly optimize the embedding vectors.

**Definition J.1.** *(Simplex ETF ([Sustik et al., 2007](#))). A set of vectors $\{\boldsymbol{\mu}_i\}_{i=1}^{C}$ in $\mathbb{R}^d$ forms a simplex Equiangular Tight Frame (ETF) if $\|\boldsymbol{\mu}_i\| = 1$ for $\forall i \in [C]$ and $\boldsymbol{\mu}_i^{\top} \boldsymbol{\mu}_j = -1/(C-1)$ for $\forall i \neq j$.*

Next, we will characterize the optimal solution for the separation loss defined as:

$$\mathcal{L}_{\text{sep}} = \frac{1}{C} \sum_{i=1}^{C} \log \underbrace{\frac{1}{C-1} \sum_{j \neq i, j=1}^{C} \exp\left(\boldsymbol{\mu}_i^{\top} \boldsymbol{\mu}_j / \tau\right)}_{\uparrow \text{ separation}} := \frac{1}{C} \sum_{i=1}^{C} \log \mathcal{L}_{\text{sep}}(i)$$

**Lemma J.1.** *(Optimal solution of the separation loss) Assume the number of classes $C \leq d+1$, $\mathcal{L}_{sep}$ is minimized when the learned class prototypes $\{\boldsymbol{\mu}_i\}_{i=1}^{C}$ form a simplex ETF.*

*Proof.*

$$\mathcal{L}_{\text{sep}}(i) = \frac{1}{C-1} \sum_{j \neq i, j=1}^{C} \exp\left(\boldsymbol{\mu}_i^{\top} \boldsymbol{\mu}_j / \tau\right) \tag{12}$$

$$\geq \exp\left(\frac{1}{C-1} \sum_{j \neq i, j=1}^{C} \boldsymbol{\mu}_i^{\top} \boldsymbol{\mu}_j / \tau\right) \tag{13}$$

$$= \exp\left(\frac{\boldsymbol{\mu}_i^{\top} \boldsymbol{\mu} - \boldsymbol{\mu}_i^{\top} \boldsymbol{\mu}_i}{\tau(C-1)}\right) \tag{14}$$

$$= \exp\left(\frac{\boldsymbol{\mu}_i^{\top} \boldsymbol{\mu} - 1}{\tau(C-1)}\right) \tag{15}$$

where we define $\boldsymbol{\mu} = \sum_{i=1}^{C} \boldsymbol{\mu}_i$ and (13) follows Jensen's inequality. Therefore, we have

$$\mathcal{L}_{\text{sep}} = \frac{1}{C} \sum_{i=1}^{C} \log \mathcal{L}_{\text{sep}}(i)$$

$$\geq \frac{1}{C} \sum_{i=1}^{C} \log \exp\left(\frac{\boldsymbol{\mu}_i^{\top} \boldsymbol{\mu} - 1}{\tau(C-1)}\right)$$

$$= \frac{1}{\tau C(C-1)} \sum_{i=1}^{C} (\boldsymbol{\mu}_i^{\top} \boldsymbol{\mu} - 1)$$

$$= \frac{1}{\tau C(C-1)} \boldsymbol{\mu}^{\top} \boldsymbol{\mu} - \frac{1}{\tau(C-1)}$$

It suffices to consider the following optimization problem,

$$\text{minimize} \quad \mathcal{L}_1 = \boldsymbol{\mu}^{\top} \boldsymbol{\mu}$$
$$\text{subject to} \quad \|\boldsymbol{\mu}_i\| = 1 \quad \forall i \in [C]$$

where $\boldsymbol{\mu}^{\top} \boldsymbol{\mu} = (\sum_{i=1}^{C} \boldsymbol{\mu}_i)^{\top} (\sum_{i=1}^{C} \boldsymbol{\mu}_i) = \sum_{i=1}^{C} \sum_{j \neq i} \boldsymbol{\mu}_i^{\top} \boldsymbol{\mu}_j + C$

However, the problem is non-convex. We first consider a convex relaxation and show that the optimal solution to the original problem is the same as the convex problem below,

$$\text{minimize} \quad \mathcal{L}_2 = \sum_{i=1}^{C} \sum_{j=1, j \neq i}^{C} \boldsymbol{\mu}_i^{T} \boldsymbol{\mu}_j$$
$$\text{subject to} \quad \|\boldsymbol{\mu}_i\| \leq 1 \quad \forall i \in [C]$$

Note that the optimal solution $\mathcal{L}_1^* \geq \mathcal{L}_2^*$. Next, we can obtain the Lagrangian form:

$$\mathcal{L}(\boldsymbol{\mu}_1, \ldots, \boldsymbol{\mu}_C, \lambda_1, \ldots, \lambda_C) = \sum_{i=1}^{C} \sum_{j=1, j \neq i}^{C} \boldsymbol{\mu}_i^T \boldsymbol{\mu}_j + \sum_{i=1}^{C} \lambda_i(\|\boldsymbol{\mu}_i\|^2 - 1)$$

where $\lambda_i$ are Lagrange multipliers. Taking the gradient of the Lagrangian with respect to $\boldsymbol{\mu}_k$ and setting it to zero, we have:

$$\frac{\partial \mathcal{L}}{\partial \boldsymbol{\mu}_k} = 2 \sum_{i \neq k}^{C} \boldsymbol{\mu}_i + 2\lambda_k \boldsymbol{\mu}_k = 0$$

Simplifying the equation, we have:

$$\boldsymbol{\mu} = \boldsymbol{\mu}_k(1 - \lambda_k)$$

Therefore, the optimal solution satisfies that (1) either all feature vectors are co-linear (*i.e.* $\boldsymbol{\mu}_k = \alpha_k \boldsymbol{v}$ for some vector $\boldsymbol{v} \in \mathbb{R}^d \ \forall k \in [C]$) or (2) the sum $\boldsymbol{\mu} = \sum_{i=1}^{C} \boldsymbol{\mu}_i = \mathbf{0}$. The Karush-Kuhn-Tucker (KKT) conditions are:

$$\boldsymbol{\mu}_k(1 - \lambda_k) = \mathbf{0} \quad \forall k$$
$$\lambda_k(\|\boldsymbol{\mu}_k\|^2 - 1) = 0 \quad \forall k$$
$$\lambda_k \geq 0 \quad \forall k$$
$$\|\boldsymbol{\mu}_k\| \leq 1 \quad \forall k$$

When the learned class prototypes $\{\boldsymbol{\mu}_i\}_{i=1}^{C}$ form a simplex ETF, $\boldsymbol{\mu}_k^\top \boldsymbol{\mu} = 1 + \sum_{i \neq k} \boldsymbol{\mu}_i^\top \boldsymbol{\mu}_k = 1 - \frac{C-1}{C-1} = 0$. Therefore, we have $\boldsymbol{\mu} = \mathbf{0}$, $\lambda_k = 1$, $\|\boldsymbol{\mu}_k\| = 1$ and KKT conditions are satisfied. Particularly, $\|\boldsymbol{\mu}_k\| = 1$ means that all vectors are on the unit hypersphere and thus the solution is also optimal for the original problem $\mathcal{L}_1$. The solution is optimal for $\mathcal{L}_{\text{sep}}$ as Jensen's inequality (13) becomes equality when $\{\boldsymbol{\mu}_i\}_{i=1}^{C}$ form a simplex ETF. The above analysis provides insights on why $\mathcal{L}_{\text{sep}}$ promotes inter-class separation.

