# OpenReview forum: "HYPO: Hyperspherical Out-Of-Distribution Generalization"
_ICLR.cc/2024/Conference — ICLR 2024 poster_

### Official Review · Reviewer_kMBq · 2023-10-22

**Soundness:** 3 good
**Presentation:** 3 good
**Contribution:** 2 fair
**Rating:** 3
**Confidence:** 5

**Summary:**

This paper delves into the challenge of out-of-distribution (OOD) generalization. Building upon previous research, it introduces the HYPO learning algorithm aimed at reducing intra-class variation while increasing inter-class separation. Notably, the paper establishes a connection between the loss function and the von Mises-Fisher (vMF) distribution. Subsequently, it provides a generalization upper bound of variation. These set HYPO apart from an existing work PCL. Extensive experimentation on OOD benchmarks showcases the superior performance of the HYPO algorithm.

**Strengths:**

- This paper is well-written and well-organized.
- The problem studied in this paper is interesting and important.
- The authors have provided a clear discussion of the relation to previous work, PCL.

**Weaknesses:**

1. The theoretical result appears to have limitations.
- Although Theorem 5.1 provides insights into the upper bound of generalization variation, it does not conclusively demonstrate the superiority of the proposed method or loss, since the theorem directly assumes that the variation can be optimized to a small value under the proposed loss, i.e., $\frac{1}{N}\sum_j\mu_{c(j)}^T z_j\ge 1-\varepsilon$. If one were to substitute an alternative loss, such as changing the prototype to another sample within the same class (e.g., employing the SupCon loss) or directly using PCL's loss, it would also yield a generalization bound. Consequently, the question arises: How can we establish that the proposed loss is indeed superior, provably?
- Theorem 5.1 cannot be valid unless we explicitly specify the distribution distance  $\rho$.
- Theorem 5.1 does not account for the influence of inter-class separation, a key aspect that this paper seeks to enhance through the second term in loss eq. (5). I notice that in Ye et al (2021)'s Theorem 4.1, function O(.) also depends on additional factors beyond just the variation.

2. Training Loss.
- Since prototypes $\mu_i$ are updated in an EMA manner, it's worth noting that the second term in eq. (5) will not generate a gradient for $h$. Consequently, the second term of the loss becomes devoid of meaning.

3. The idea is quite straightforward and shares many similarities with proxy-based contrastive learning methods. Is there any additional insight that I might have overlooked?

4. The empirical improvements appear to be marginal, as indicated by the data in Tables 1 and 2.

Overall, I think the theoretical contribution and empirical enhancements appear to have room for further development and strengthening.

**Questions:**

See weaknesses.

---

> ### Author Response · Authors · 2023-11-16
> **Response to Reviewer kMBq**
>
> We sincerely appreciate your comments and questions, which we address in detail below.
>
> > *W1.1. Although Thm 5.1 provides insights into the upper bound of generalization variation, it does not conclusively demonstrate the superiority of the proposed method or loss.*
>
> We would like to clarify that our goal is not to theoretically establish HYPO's superiority over SupCon and PCL. Our primary contribution is to provide the **first theoretical justification** of prototype-based learning algorithms for OOD generalization. This is also acknowledged by Reviewer W1ps, who commented:
>
> > _"...several learning methods have previously been proposed that utilize hyperspherical embeddings. But this paper is the first to provide a theoretical justification angled at OOD generalization."_
>
> Unlike PCL, our learning framework HYPO is motivated and guided by theory (Section 3.1), and provably guaranteed to reduce OOD generalization error (Thm 3.1 & Thm 5.1). Indeed, there may exist several alternative proxy-based learning algorithms such as PCL. However, it remains unclear if PCL's performance can be rigorously interpreted and bounded. We believe that our theoretical framework takes an important first step towards understanding domain generalization via hyperspherical learning formally, paving the way for more effective and theoretically grounded approaches in the field.
>
> Empirically, unlike PCL which relies on SWAD optimization, HYPO is able to achieve strong performance by simple SGD optimization and the performance can be further improved with SWAD. Compared to PCL, HYPO achieves superior performance of an average accuracy of 89% on PACS, further demonstrating its effectiveness in practice.
>
> > *W1.2. Thm 5.1 cannot be valid unless we explicitly specify the distribution distance $\rho$.*
>
> Indeed! We specified the distribution distance $\rho$ as shown in **Definition 3.1** in the main manuscript. We have also added a footnote in the updated main manuscript for clarity.
>
> > *W1.3. Thm 5.1 does not account for the influence of inter-class separation. Besides, in Ye et al (2021)'s Thm 4.1, function O(.) also depends on additional factors beyond just the variation.*
>
> We address the effect of inter-class separation in our theoretical analysis presented in **Appendix J**, which links to the second term in our loss eq. (5). Additionally, for the discussion on regularity conditions in Thm 4.1, please see our response to Reviewer W1ps.
>
> > *W2. Clarification on the prototype update rule via EMA and the meaning of the second term in the loss.*
>
> The separation term's primary purpose is to maintain distinctiveness between class prototypes in the embedding space. While the separation term may not directly affect the gradients of $\mathbf{z}$, it indirectly influences how $\mathbf{z}$ is learned. By keeping the prototypes separate, the variation term has to adapt $\mathbf{z}$ in a way that aligns with these well-separated prototypes, leading to better OOD generalization. As detailed in Algorithm 1 in Appendix A, in this work, we choose to update the prototypes $\boldsymbol{\mu}_i$ via EMA, which leads to stable updates of prototypes. Alternatively, one can set class prototypes as learnable parameters and update prototypes via gradients. We compare the performance of our method with EMA update and learnable prototypes (LP) in Appendix H (P24). We observe our method with EMA achieves superior performance compared to LP, which empirically verifies the effectiveness of EMA-style update of prototypes in practice.

---

> > ### Author Response · Authors · 2023-11-16
> > **Response to Reviewer kMBq (cont.)**
> >
> > > *W3. The idea is quite straightforward and shares many similarities with proxy-based contrastive learning methods. Is there any additional insight that I might have overlooked?*
> >
> >
> > We provide a comparison in the paragraph *Relation to PCL [1]* on page 7. Here we would like to highlight notable distinctions and provide further clarifications:
> >
> > - [**Theoretical grounding**] PCL offers no theoretical insights, while our loss HYPO is guided by theory. We provide a formal theoretical justification that our method reduces intra-class variation which is essential to bounding OOD generalization error (Section 5).
> > - [**Statistical interpretation via vMF**] While PCL and our loss share high-level intuitions, our loss formulation can be rigorously interpreted as shaping vMF distributions of hyperspherical embeddings (Section 4.2). In contrast, it remains unclear if PCL can be statistically interpreted.
> > - [**Inter-class separation**] HYPO explicitly promotes inter-class separation via the prototype-to-prototype loss, whereas PCL does not. This loss is directly motivated by and grounded in the theoretical framework by Ye et al.
> > - [**Strong performance w/o specialized optimization technique SWAD**] Unlike PCL which relies on SWAD, a dense and overfit-aware stochastic weight sampling strategy for OOD generalization, HYPO is able to achieve strong performance by simple SGD optimization (Avg Acc 88.0\% without SWAD) and the performance can be further improved with SWAD.
> >
> > > *W4. Clarification on empirical improvements*
> >
> > We appreciate Reviewer kMBq's scrutiny of our empirical results. We believe, however, that the improvements we demonstrate are both statistically significant and practically meaningful, particularly when considering the challenging nature of the OOD generalization problem and the competitive landscape of the field. As Reviewer W1ps commended, our empirical evaluation is extensive, where HYPO consistently outperforms a wide range of baseline methods on challenging OOD benchmarks. In particular, HYPO demonstrates an overall 1.2% improvement on average (Table 1), which is substantial compared to the relative improvement of competitive baselines. This achievement is also underscored by the 89% accuracy (w. SWAD) reported in Table 2, reflecting a notable advance in a highly competitive area.
> >
> > In light of HYPO's consistently demonstrated superiority, its straightforward and efficient implementation, and theoretical foundations, we believe that HYPO represents a notable advancement in the field.

---

> ### Comment · Reviewer_kMBq · 2023-11-19
> **Thanks for the Detailed Rebuttal**
>
> Thank you for your comprehensive and detailed rebuttal. I have carefully read through your responses to the comments and truly appreciate the time and effort you have put into addressing each concern. My further comments are listed as follows.
>
> ---
>
> (1) This paper claims at the beginning of section 5 that
> > Our main Theorem 5.1 gives a
> provable understanding of **how the learning objective effectively reduces the variation estimate**.
>
> However, Theorem 5.1 presupposes that the variation can be optimized to a small value under the proposed loss, specifically $\frac{1}{N}\sum_j\mu_{c(j)}^T z_j\ge 1-\varepsilon$.
> Notably, the paper lacks a clear explanation of how the learning objective in Eq (5) upper bounds $1-\frac{1}{N}\sum_j\mu_{c(j)}^T z_j$, which is crucial for justifying the claimed reduction in variation.
> Even for a special and simpler case where infinity samples are accessible $N\rightarrow\infty$, the theorem can still not tell why the learning objective is able to effectively reduce the variation estimate.
>
> Therefore,
> the theoretical justification of prototype-based learning algorithms for OOD generalization is not sufficiently informative.
>
> Moreover, it seems under this assumption, PCL and other methods can  also offer the same theoretical insights.
>
> ---
>
>
> (2) Why is there only a footnote directing to Appendix C for the theorem in the revised paper? Firstly, while Appendix C is extensive, it would be more helpful if the authors could provide a more specific reference within it for easier navigation. Additionally, why do the authors not directly include the distance $\rho$ used in Definition 3.1, as metentioned in the rebuttal?
>
> Secondly, for the sake of rigor, it is essential for the theorem to be self-contained. Including specific details, such as the distance $\rho$ from Definition 3.1, within the theorem would enhance clarity and completeness.
>
> ---
>
> (3) I appreciate the effort made to demonstrate that a simplex ETF can minimize the separation part of the proposed loss. However, my initial concern remains unanswered: how does the separation part of the proposed loss impact the OOD error?
>
>
> ---
>
> (4) I cannot understand how the separation term can indirectly influence $\mathbf{z}$ if it does not directly affect the gradients of $\mathbf{z}$, as the authors mentioned in the rebuttal. A more detailed and mathematical explanation would be appreciated to clarify this aspect.
>
> Moreover, my initial concern revolves around the potential redundancy of the separation term when EMA is employed. In the rebuttal, the experiments compare the loss with EMA and without EMA, which certainly introduces a distinction between the two settings. To address this concern directly, it would be beneficial to conduct experiments comparing the loss with the separation term and without the separation term specifically when employing EMA.
>
> ---
>
> The manuscript would benefit from further refinement to fully meet the standards of ICLR. Thus, I believe this submission will be a good paper, but for now, it is not ready. I have decided to maintain my original score, and thank you for your effort once again.

---

> > ### Author Response · Authors · 2023-11-20
> > **Response to Reviewer kMBq (Follow-up)**
> >
> > We appreciate your follow up questions and comments. We would like to clarify these points further.
> >
> > > (1) How does learning objective in Eq (5) bound $\frac{1}{N} \sum\_{j=1}^N \boldsymbol{\mu}\_{c(j)}^{\top} \mathbf{z}\_j \geq 1-\epsilon$?
> >
> > To see this, we refer reviewer to our variation loss term $\mathcal{L}_\text{var}$
> > $$
> >   \mathcal{L}\_\text{var} =   - \frac{1}{N} \sum\_{e \in \mathcal{E}\_\text{avail}} \sum\_{i=1}^{|\mathcal{D}^e|} \log \frac{\exp \left({\mathbf{z}^{e}\_{i}}^\top  {\boldsymbol{\mu}}\_{c(i)} / \tau\right)}{\sum\_{j=1}^{C} \exp \left({\mathbf{z}^{e}\_{i}}^\top  {\boldsymbol{\mu}}\_{j} / \tau\right)}
> > $$
> >
> > $$
> > = - \frac{1}{N} \sum\_{e \in \mathcal{E}\_\text{avail}} \sum\_{i=1}^{|\mathcal{D}^e|} {\mathbf{z}^{e}\_{i}}^\top  {\boldsymbol{\mu}}\_{c(i)} / \tau + \frac{1}{N} \sum\_{e \in \mathcal{E}\_\text{avail}} \sum\_{i=1}^{|\mathcal{D}^e|} \log \sum\_{j=1}^{C} \exp \left({\mathbf{z}^{e}\_{i}}^\top  {\boldsymbol{\mu}}\_{j} / \tau\right)
> > $$
> >
> > $$
> > = -\frac{1}{N \tau} \sum\_{j=1}^N \boldsymbol{\mu}\_{c(j)}^{\top} \mathbf{z}\_j + \frac{1}{N} \sum\_{e \in \mathcal{E}\_\text{avail}} \sum\_{i=1}^{|\mathcal{D}^e|} \log \sum\_{j=1}^{C} \exp \left({\mathbf{z}^{e}\_{i}}^\top  {\boldsymbol{\mu}}\_{j} / \tau\right),
> > $$
> > where **minimizing our variation loss $\mathcal{L}_\text{var}$ is equivalent to maximizing $\frac{1}{N} \sum\_{j=1}^N \boldsymbol{\mu}\_{c(j)}^{\top} \mathbf{z}\_j \rightarrow 1$** in the first term. Under sufficient expressivity of neural networks, the optimization error is small, which directly translate into small $\epsilon$ (empirical verification can be seen in Appendix H and Table 14).
> >
> >
> > > (2) Why is there only a footnote directing to Appendix C for the theorem in the revised paper? Firstly, while Appendix C is extensive, it would be more helpful if the authors could provide a more specific reference within it for easier navigation. Additionally, why do the authors not directly include the distance used in Definition 3.1, as metentioned in the rebuttal?
> >
> > Thanks for pointing out the typo -- we should have directed it to Definition 3.1 instead. This has been fixed in our updated manuscript.
> >
> > On a separate note, the distance metrics we choose in this work are well-established concepts in the field (e.g., Larry Wasserman's notes: https://www.stat.cmu.edu/~larry/=sml/Opt.pdf). We believe that focusing on more central aspects of our work, without overextending on relatively well-understood concepts, makes for a clearer and more concise presentation. This approach also helps us to adhere to space limitations.
> >
> > > (3.1) I appreciate the effort made to demonstrate that a simplex ETF can minimize the separation part of the proposed loss.
> >
> > Thank you for acknowledging our efforts in analyzing the simplex ETF and its role in the separation part of our loss function. The simplex ETF configuration leads to class prototypes equally spaced on the unit hypersphere, thereby maximizing inter-class separation.
> >
> >
> > > (3.2) However, my initial concern remains unanswered: how does the separation part of the proposed loss impact the OOD error?
> >
> > **Please kindly note that addressing this question is the contribution made in Ye et al., whereas our new contribution and focus is to design the loss that achieves this separation property** (for which we rigorously proved in Appendix J).  For completeness, we have provided a summary in Appendix C on how separation impact OOD error. If confusion remains, we refer reviewer to Ye et al. for a detailed answer to this question with more extensive explanations: the definition of inter-class separation ('informativeness') provided in Def 3.2 (P5), the necessity of informativeness (P6), the relation between informativeness and OOD generalization in Thm 4.1 (P7) in Ye et al.

---

> > > ### Author Response · Authors · 2023-11-20
> > > **Response to Reviewer kMBq (cont.)**
> > >
> > > > (4) I cannot understand how the separation term can indirectly influence z. if it does not directly affect the gradients of z, as the authors mentioned in the rebuttal. A more detailed and mathematical explanation would be appreciated to clarify this aspect.
> > >
> > > > Moreover, my initial concern revolves around the potential redundancy of the separation term when EMA is employed. In the rebuttal, the experiments compare the loss with EMA and without EMA, which certainly introduces a distinction between the two settings. To address this concern directly, it would be beneficial to conduct experiments comparing the loss with the separation term and without the separation term specifically when employing EMA.
> > >
> > > **[A detailed explanation on the impact of separation term and whether loss is redundant]**
> > >
> > > We would like to provide further detailed and mathematical explanations as follows. Recall that the class prototype update rule (Eq. 6 in P4)
> > > $$
> > > \boldsymbol{\mu}\_c := \operatorname{Normalize}\left(\alpha \boldsymbol{\mu}\_c + (1 - \alpha) \mathbf{z}\right), \forall c \in \{1, 2, \ldots, C\}
> > > $$
> > > Here, the normalized feature $\mathbf{z}=\tilde{\mathbf{z}} /\left\\|\tilde{\mathbf{z}}\right\\|\_2$ where $\tilde{\mathbf{z}}=h_{\mathbf{w}}\left(\tilde{\mathbf{x}}\right)$ (see Algorithm 1 in P15) is a function of the neural network weights (denoted as $\mathbf{w}$). According to the chain rule, $\mathcal{L} \rightarrow \boldsymbol{\mu} \rightarrow \mathbf{z} \rightarrow \mathbf{w}$, the neural network weights are updated by the gradient of the loss function, which includes the separation term.
> > >
> > > Given this, we can see that when the loss function guides the update of the network weights, it also influences the features $\mathbf{z}$ extracted by the network. These feature updates, in turn, influence the class prototypes $\boldsymbol{\mu}\_c$ via the EMA mechanism. Therefore, while the prototypes themselves are not directly optimized by the loss gradient under the EMA scheme, they are updated in a manner that reflects the changes in the network induced by the loss.
> > >
> > > **[Empirical comparison with and without the separation term when employing EMA]** To further verify the effectiveness of the separation term *when employing EMA*, we **have provided an ablation study in Appendix H (p23)**, where we compare the OOD generalization performance of our method (with separation loss) vs. our method (without separation loss) under when employing EMA on PACS. As shown in **Table 11**, we observe that including the separtion term leads to superior performance compared to that without the separation term.

---

> ### Comment · Reviewer_kMBq · 2023-11-20
> **Maintain My Original Score**
>
> Dear Authors,
>
> I have reviewed your latest feedback, including the appendix, and I want to express my gratitude for your diligent response. My further comments are as follows.
>
> (1) This paper highlights its theoretical significance as a main contribution (page 2, also the paper title), **claiming to provide theoretical justification for how the proposed loss guarantees improved OOD generalization**.
> Consequently, I expected a direct theorem demonstrating that **the OOD error can be bounded by the proposed loss**. However, this paper fails to provide this crucial argument, which largely limits its novelty.
>
> (2) The rebuttal claims that minimizing **the variation loss** is "equivalent" to maximizing the negative **first term of variation loss**. This claim seems to lack sufficient mathematical rigor.
>
> (3) In fact, the proposed loss comprises three components: `L = first term of variation loss + second term of variation + separation loss`.
> While minimizing the first term of the variation alone ensures that $\frac{1}{N}\sum_j\mu_{c(j)}^T z_j$ is close to 1, **the presence of the last two parts tends to deviate it from 1**. Therefore, **the gap between $\frac{1}{N}\sum_j\mu_{c(j)}^T z_j$ and one should be the most important part of this paper** and needs to be thoroughly investigated, instead of assuming its proximity to 1 directly (while this paper also claims that the introduction of separation loss is essential).
>
> (4) Otherwise, if one uses only the first term of variation loss as the total loss, according to your theorem, the ood performance should also be good or even better (since smaller $\varepsilon$), which contradicts observed facts. This challenges the applicability of the theorem.
>
> (5) Furthermore, based on your assumption, it appears that PCL and other methods could potentially provide similar theoretical insights, thereby largely diminishing the contribution of your approach compared to PCL.
>
> (6) Similarly, since you claim that the proposed loss guarantees improved OOD generalization, you should point out **how the separation loss affects the ood error in your theorem** (the necessity of the separation loss). Otherwise, why can't one remove the separation loss? This concern does not appear to be adequately addressed in Appendix J.
>
> (7) About EMA: The proposed loss can be written as $L(z,\mu)=variation(z,\mu)+separation(\mu)$. Since $\mu$ is updated by EMA, it is fixed when computating the gradients, i.e., $\nabla_z L(z,\mu)=\nabla_z variation(z,\mu)+\nabla_z separation(\mu)=\nabla_z variation(z,\mu)$. Then the encoder updates its weight based on $\nabla_z variation(z,\mu)$. After that, $\mu$ is updated by EMA, depending on the previous $\mu$ and current $z$, not depending on the loss. Therefore, the separation loss is not used in the training procedure.
>
>
> ---
>
> In summary, this paper appears to overstate its **theoretical** contributions and demonstrates only marginal **empirical** improvements, possibly with technical flaws. Consequently, I am inclined to reject it.

---

> > ### Author Response · Authors · 2023-11-23
> > **Response to Reviewer kMBq (Follow-up)**
> >
> > We sincerely appreciate your time and effort in following up. While we acknowledge and respect varying opinions in the peer-review process, we wish to express our concerns below.
> >
> > In the spirit of fostering a constructive conversation, we believe that a thoughtful examination of the work and author responses are essential during the author-reviewer interaction period.
> > If there are lingering points of confusion, we encourage the reviewer to frame them as questions rather than **biased assertions** to foster more open conversations.
> >
> > We believe that through open-minded dialogue, we can foster an environment conducive to fair evaluations and constructive feedback.
> >
> > We would like to take the opportunity and provide further clarifications:
> >
> > > (1) This paper highlights its theoretical significance as a main contribution (page 2, also the paper title), claiming to provide theoretical justification for how the proposed loss guarantees improved OOD generalization. Consequently, I expected a direct theorem demonstrating that the OOD error can be bounded by the proposed loss. However, this paper fails to provide this crucial argument, which largely limits its novelty.
> >
> > **We respectfully disagree with this falsified critism**. The formal guarantee of how our proposed loss improves OOD generalization has been provided directly in **Theorem 5.1** (along with full proof in Appendix). Contrary to what you said, our key contribution is introducing such theoretical justification for prototype-based learning in OOD generalization. Previous studies have not fully explored the theoretical aspects of OOD generalization and algorithmic guarantees. Our framework represents a significant step in formally understanding OOD generalization through hyperspherical learning.
> >
> > > (2) The rebuttal claims that minimizing the variation loss is "equivalent" to maximizing the negative first term of variation loss. This claim seems to lack sufficient mathematical rigor.
> >
> > **It appears there has been a misunderstanding regarding our original statement**. We intended to say minimizing the first term in our variation loss is equivalent to maximizing $\frac{1}{N} \sum_{j=1}^N \boldsymbol{\mu}_{c(j)}^{\top} \mathbf{z}_j$.
> >
> > > (3) While minimizing the first term of the variation alone ensures that $\frac{1}{N} \sum\_{j=1}^N \boldsymbol{\mu}\_{c(j)}^{\top} \mathbf{z}\_j$ is close to 1, the presence of the last two parts tends to deviate it from 1. The gap between $\frac{1}{N} \sum\_{j=1}^N \boldsymbol{\mu}\_{c(j)}^{\top} \mathbf{z}\_j$ and one should be the most important part of this paper and needs to be thoroughly investigated, instead of assuming its proximity to 1 directly.
> >
> > **We respectfully disagree with the misinterpretation that the last two components cause the first term of variation loss to deviate from 1**. The first term of variation is applied within the same class (promote low intra-class variation), whereas the last two terms target different classes (promote high inter-class separation). For empirical evidence, refer to the results of $\hat{\epsilon} := 1 - \frac{1}{N} \sum\_{j=1}^N \boldsymbol{\mu}\_{c(j)}^{\top} \mathbf{z}\_j$ when trained with HYPO, as detailed in Appendix H. **Our investigation provides concrete evidence that training with HYPO significantly reduces the average intra-class variation, leading to $\frac{1}{N} \sum\_{j=1}^N \boldsymbol{\mu}\_{c(j)}^{\top} \mathbf{z}\_j$ being close to 1 in practice**.

---

> > > ### Author Response · Authors · 2023-11-23
> > > **Response to Reviewer kMBq (cont.)**
> > >
> > > > (4) If one uses only the first term of variation loss as the total loss, according to your theorem, the ood performance should also be good or even better, which contradicts observed facts.
> > >
> > > **We respectfully disagree with the ungrounded assertion that using only the first term of variation loss would yield better results**. We would like to provide clarifications below to avoid the misinterpretaion of our theory.
> > >
> > > The second term of variation loss contributes to the uniformity (informativeness property). Both low intra-class variation and high inter-class separation are desirable properties for theoretically grounded OOD generalization. This is based on the theoretical framework proposed by Ye et al., which incorporates inter-class separation into the learnability aspect of the OOD generalization problem.
> > >
> > > In summary, when the learned embeddings exhibit high inter-class separation, the problem becomes learnable. In this context, bounding intra-class variation becomes crucial for reducing the OOD generalization error. We provide a brief summary in Appendix C to discuss the notion of OOD learnability, and would like to refer readers to Ye et al. for an in-depth and formal treatment.
> > >
> > > Empirically, to verify the impact of inter-class separation, we conducted an ablation study detailed in **Appendix H** where we compare the OOD performance of our method (with separation loss) vs. our method (without separation loss). We observe that incorporating separation loss indeed achieves stronger OOD generalization performance.
> > >
> > > | Method | OOD Acc. (%) |
> > > | -------| ------- |
> > > | HYPO (w/o separation loss) | 87.2|
> > > | HYPO (w separation loss)   | **88.0**|
> > >
> > > > (5) PCL and other methods could potentially provide similar theoretical insights.
> > >
> > > It's important to clarify that our goal is not to demonstrate HYPO's theoretical superiority over PCL or other methods. Instead, our key contribution is to provide the first theoretical justification of prototype-based learning algorithms for enhancing OOD generalization. No prior works have provided theoretical justifications on how prototype-based methods improve OOD generalization. **While PCL and other prior methods *could potentially provide* similar theoretical insights, investigating the theoretical property of PCL is outside the scope of our work**.
> > >
> > > > (6) How the separation loss affects the ood error in your theorem (the necessity of the separation loss).
> > >
> > > **We would like to direct your attention again to our previous reply to your question (4)**. Our theoretical framework, based on the work of Ye et al., incorporates inter-class separation into the learnability aspect of OOD generalization. Essentially, when learned embeddings display substantial inter-class separation, the OOD generalization problem becomes learnable. In this scenario, limiting intra-class variation is pivotal for reducing OOD generalization error. We offer a concise summary in Appendix C on the concept of OOD learnability and encourage readers to consult the detailed discussion in Ye et al.'s work for further understanding.
> > >
> > > > (7) Utilization of separation loss in training.
> > >
> > > The primary role of the separation term is to ensure distinctiveness between class prototypes in the embedding space, indirectly influencing the learning of $\mathbf{z}$. We advise a **careful review** of line 8 in Algorithm 1 (Appendix A) and the definition of EMA, illustrating that $\boldsymbol{\mu}$ is functionally dependent on $\mathbf{z}$.

---

### Official Review · Reviewer_W1ps · 2023-11-01

**Soundness:** 4 excellent
**Presentation:** 3 good
**Contribution:** 3 good
**Rating:** 6
**Confidence:** 4

**Summary:**

This paper introduces a pracical algorithm for achieving provable out-of-distribution (OOD) generalization. The proposed approach is motivated by recent theoretical work that decomposes OOD generalization into two measurable quantities: intra-class variation and inter-class separation. This paper designs a training objective (and representation space) where these terms can be optimized to achieve low OOD generalization error.

Specifically, the proposed method learns representations for each data point that lie on a hypersphere. The goal is to encourage data points belonging to the same class to lie close together on the hypersphere (in terms of cosine distance) but to have the centroids of each class lie far apart. This approach itself is not particularly novel, as several learning methods have previously been proposed that utilize hyperspherical embeddings. But this paper is the first to provide a theoretical justification angled at OOD generalization.

The paper provides a formal theoretical proof that bounds the OOD generalization error via a standard PAC-like learning bound. The proof leans on the prior theoretical results that motivated this work.

**Strengths:**

This paper proposed a simple algorithm that is easy to implement. The loss terms can be computed efficiently and are easy to mini-batch for SGD. The authors provide a clear description of the algorithm and even include pseudo-code. It would be easy to reproduce the proposed method.

The paper is well-written and easy to follow. Motivation is laid out clearly and the paper accurately describes its contributions relative to prior work. I was able to find all of the information that I wanted while reading the paper either within the main text or the appendices.

I see the primary contribution of this work to be the formal theoretical guarantee on the generalization performance of the proposed method. The theoretical results presented in this work are environment agnostic in the sense that they only depend on the environments through the ability to fit the training data effectively and reduce the intra-class variation. This is a valuable contribution.

The empirical results are relatively thorough and compare HYPO (the proposed method) against a wide range of baseline methods across several tasks. The results show that HYPO performs well consistently, and is on average the best OOD classifier.

I liked the simple theoretical exploration in Appendix J. This was a valuable inclusion that helped to give some intuition for the class separation loss component.

**Weaknesses:**

The paper lacks quantitative verification of the theoretical result. I think that this would be a valuable contribution to help give an idea of how tight/vacuous the bound is. I am mostly curious about the $\epsilon$ term that appears in Theorem 5.1 and can be easily computed in practice.

The theoretical result shown gives a bound on the intra-class variation. This is a useful component of producing an OOD generalization bound, but it is not sufficient by itself. The results in Ye et al. require some regularity conditions that depend on the distribution over the learned representations --- this is difficult to compute in this case. From my point of view, the theoretical results in this paper provide a strong intuition for the success of the method but have not yet been demonstrated to produce a tractable OOD generalization bound.

Spurious correlations are ignored in this work, though are one of the more challenging aspects of OOD generalization in practice. However, I think that this is a reasonable compromise to make at this stage.

I feel that the novelty is slightly limited here. The proposed learning algorithm is a form of prototypical learning on a hypersphere. The specific loss is, to my knowledge, novel but is made up of fairly standard components. The theoretical results are novel and interesting, but are essentially an instantiation of results from prior work. Indeed, the contribution of the training loss to the generalization error is largely captured in an assumption within the theoretical statement. I do consider the overall novelty of this paper to be sufficient for me to recommend acceptance, but it has affected my overall judgment so I am including this as a weakness.

**Questions:**

- I'd appreciate it if the authors could explain the motivation behind Equation 6 a little more.  Is the primary goal to improve on the computational efficiency of computing the average across all training data points? Or is there another benefit to adopting an exponential moving average? This also ties loosely into my next question.

- How strong is the assumption that the samples are aligned? Intuitively, the intra-class variation measures how much the features vary across environments for a single class. The alignment assumption is an assumption over all of the training data in the available environments. Consequently, Theorem 5.1 consists of a term that depends on epsilon, and a generalization term that (intuitively) describes generalization to the unavailable environments. I think it would be more valuable if one could show that the alignment assumption is satisfied by reducing the training loss directly, bringing the result more in line with typical PAC generalization bounds.

- The epsilon factor could potentially make the bound very loose if it is too close to 1. Given that this value is easy to compute, I would be curious to know what epsilon looks like for some of the models trained in the experiments.

- Ye et al. provide a specialized result for linear models (Theorem 4.2 in their work). I see this as a justification that the theoretical framework can be realized by some model. However, in the present work, it is unclear whether the vMF distribution can satisfy the regularity conditions for some choice of environment distribution(s). In other words, how do we know that the OOD generalization bounds can actually be computed for the choice of model used?


Minor comments:

- In the introduction, I'd recommend replacing the four lines of citations with a survey paper, for example [1]. The full list of references could be included in the related work, or even as an extended discussion of related work in the supplementary material.
- [2] is another reference that explores a contrastive metric learning approach for hyperspherical embeddings. The goal here is not to do OOD generalization, but the algorithm is modestly similar.
- In proof of Theorem 5.1, "at last $1 - \delta$" -> "at least $1-\delta$".
- It would be nice if Table 1 were sorted by ascending average accuracy.


[1]: Domain Generalization: A Survey, Zhou et al.
[2]: Video Face Clustering with Unknown Number of Clusters, Tapaswi et al. ICCV 2019

---

> ### Author Response · Authors · 2023-11-16
> **Response to Reviewer W1ps**
>
> We are grateful for your insightful comments and appreciation of our work. We address each question in detail and provide further clarifications below.
>
>
> > *Q1. The motivation behind Equation 6*
>
> As you concur, we choose to update the class prototypes using exponential-moving-average (EMA) due to computational efficiency. Otherwise, computing the average over the entire dataset would have been very expensive. This practice has been commonly used in prior literature, see for example [3].
>
>
> > *C1. Clarification on the significance of Thm 5.1*
>
> We would like to clarify that in Thm 5.1, the impact of sample-to-prototype alignment on the intra-class variation can be explicitly characterized via the first term $ε^{1\over 3}$ in the upper bound.  In particular, $ε^{1\over 3}$ measures how sample embeddings are aligned with their class prototypes on the hyperspherical space (as we have $\frac{1}{N} \sum_{j=1}^N \boldsymbol{\mu}_{c(j)}^{\top} \mathbf{z}_j \geq 1-\epsilon$), which is optimized by our proposed loss.  This Theorem implies that improved alignment (an illustration is shown in Figure 4(b)) leads to smaller upper bound of the intra-class variation $\mathcal{V}^{\text{sup}}$, a key term to upper bound the OOD generalization error by Thm 3.1. We provide empirical verifications below which further verify that the core assumption in our theoretical framework is satisfied in practice via our loss.
>
> >  Q2 Empirical verification of Thm 5.1 and $\epsilon$ factor.
>
>
> - **Quantitative verification**: Following the suggestion, we calculate the average intra-class variation over data from all environments $\frac{1}{N} \sum_{j=1}^N \boldsymbol{\mu}\_{c(j)}^{\top} \mathbf{z}\_j$ (Thm 5.1) models trained with HYPO. Then we obtain $\hat{\epsilon} := 1 - \frac{1}{N} \sum_{j=1}^N \boldsymbol{\mu}\_{c(j)}^{\top} \mathbf{z}\_j$. Due to the constraint of time, we evaluated on PACS, VLCS, and OfficeHome and summarize the results in the table below. We observe that training with HYPO significantly reduces the average intra-class variation, resulting in a small epsilon ($\hat{\epsilon} < 0.1$) in practice. This suggests that the first term $O(\epsilon^{1\over 3})$ in Thm 5.1 is indeed small for models trained with HYPO.
>
> | Dataset |$\hat{\epsilon}$ |
> |-----|-----|
> | PACS | 0.06|
> | VLCS| 0.08|
> | OfficeHome| 0.09 |
>
>
> - **Qualitative verification**: We visualized the feature representations via UMAP in Figure 4. We can see that representations learned by HYPO become significantly more aligned with class prototypes across environments. This alignment further confirms the low variation in our learned representations, which is in close agreement with the regularity conditions on the density function required for the main theorem in Ye et al.
>
>
> > *Q3. Ye et al. provide a specialized result for linear models (Theorem 4.2 in their work). I see this as a justification that the theoretical framework can be realized by some model...How do we know that the OOD generalization bounds can actually be computed for the choice of model used?*
>
> Great question! As specified in P9, we perform classification by identifying the closest class prototype: $\hat y = \text{argmax}_{c \in[C]} f_c$ where $f_c= \mathbf{z}^\top\boldsymbol{\mu}_c$ for embedding $\mathbf{z}$.  This is equivalent to the linear top model as in Ye et al. (without the bias term), as we can reformulate as $f_c= (W\mathbf{z})_c$, where the $c$-th row of $W$ is $\boldsymbol{\mu}_c$. In this case, we have a linear convergence rate without the regularity conditions in Thm 4.1 (as shown in P7 in Ye et al.). In a more general case, the generalization bound in Thm 4.1 (pasted below for easy reference) depends on additional factors
> $$\operatorname{err}(f) \leq O\left(s\left(\mathcal{V}\_\rho^{\text {sup }}\left(h, \mathcal{E}\_{\text {avail }}\right)\right)^{\frac{\alpha^2}{(\alpha+d)^2}}\right) $$
>
> such as $\alpha$ and $d$, where $d$ denotes the dimension of feature embeddings, and $\alpha$ is closely correlated to the density concentration. Such factors mainly affect the convergence rate of the error upper bound. The Theorem shows that, for any model, the generalization gap depends largely on the model’s variation captured by $\mathcal{V}^{\text {sup}}$. As HYPO aims to reduce the intra-class variation (verified in the previous response), it effectively reduces the OOD generalization bound.

---

> > ### Author Response · Authors · 2023-11-16
> > **Response to Reviewer W1ps (cont.)**
> >
> > ### Other comments:
> >
> >
> > > *C1. Typos and citations*
> >
> > Thanks for pointing out! We have fixed the typo in proof of Thm 5.1 from "at last" to "at least" and have added citation for [2] in the updated version.
> >
> > > *C2. Replacing the four lines of citations with a survey paper [1] in the Introduction*
> >
> > Thanks for the suggestion! We will replace these with the survey paper[1] to improve readability.
> >
> > > *C3. It would be nice if Table 1 were sorted by ascending average accuracy*
> >
> > Thanks for the suggestion! We have adjusted the order of algorithms in Table 1 for clarity and ease of comparison.
> >
> >
> > **References**
> >
> > [1] Zhou et al., Domain Generalization: A Survey, TPAMI 2022.
> >
> > [2] Tapaswi et al., Video Face Clustering with Unknown Number of Clusters, ICCV 2019.
> >
> > [3] Li et al., MoPro: Webly Supervised Learning with Momentum Prototypes. ICLR 2020.

---

> ### Comment · Reviewer_W1ps · 2023-11-22
>
> Thank you for your detailed response.
>
> I have taken the time to read through the discussions with other reviewers, in addition to your response here.
>
> I appreciate the additional experiments that you've included. I feel that this helps to solidify the relevance of Theorem 5.1. The $\hat{\epsilon}$ values reported lead to an approximate additive error of $\epsilon^{1/3} \approx 0.4$ for the datasets explored. I'm having a hard time qualifying how large this value is relative to the loss bound (`B`) as I couldn't easily see how this boundedness assumption impacts the generalization error upper bound. It would also help to compare this value of $\epsilon$ achieved for other baseline methods --- to verify that an improvement is obtained from the proposed method.
>
> I also share some of the concerns of kMBq, which I discussed in my original review: "I think it would be more valuable if one could show that the alignment assumption is satisfied by reducing the training loss directly, bringing the result more in line with typical PAC generalization bounds." I tend to agree with their final conclusion that this hasn't been adequately demonstrated; though the analysis you gave in response is a good start and I'd recommend that this be included.
>
> Overall, I feel that the paper has been improved in rebuttal. The addition of new empirical results helps to support the claims of the paper. I believe that the theoretical contribution could be made more complete, but I think on balance there are some good contributions in this work. While I broadly agree with Reviewer kMBq, I have chosen to maintain my current score.

---

> > ### Author Response · Authors · 2023-11-23
> > **Response to Reviewer W1ps (Followup)**
> >
> > Thank you for your insightful feedback and constructive comments, which have been invaluable in enhancing our manuscript. We will ensure to incorporate the additional results and discussions in the paper as suggested.

---

### Official Review · Reviewer_HW2k · 2023-11-04

**Soundness:** 3 good
**Presentation:** 3 good
**Contribution:** 3 good
**Rating:** 6
**Confidence:** 3

**Summary:**

This paper proposes a new loss function tailored to the out-of-distribution problem, where generalisation of the algorithm is required across multiple (and sometimes unseen) environments. Inspired by prior theoretical work, the authors devise an algorithm that encourages samples with the same label to be learnt by features that are as stable as possible across environments, while at the same time encouraging embeddings to look very dissimilar for data points with different labels. They achieve this by embedding points on the sphere and introducing class centroids per label (shared among environments), where points are encouraged to lie close to their corresponding centroid, and centroids themselves are pushed apart. The authors derive a theoretical guarantee for their algorithm and demonstrate its empirical success on CIFAR10-C, PACS and similar.

**Strengths:**

1. The paper is very well-written which made it (mostly) easy to follow as well as a pleasure to read for me.
2. The suggested loss function is very intuitive and I like the geometric interpretation the authors provide in terms of the Mises-Fisher model. The visualisation in Fig 4 is also very neat. Empirical performance is also very strong across the different explored tasks.

**Weaknesses:**

1. I struggle to see how Theorem 5.1 connects back to the proposed loss function. From Theorem 3.1 we know that $\nu^{\text{sup}}$ serves as an upper bound to the OOD error, and then Theorem 5.1 in-turn provides an upper bound for $\nu^{\text{sup}}$ in terms of the  Rademacher complexity and some additive constants. Which term here is the loss trying to minimise here? The Rademacher complexity is over any $\sigma_i$, so its sign has nothing to do with the true labels. I don’t see how the developed loss would encourage to minimise this quantity. It’s also a worst-case bound in terms of the hypothesis $h$, so again I don’t see how that could be minimised. I hope the authors can elaborate on this connection.
2. The CIFAR10-C results look strong but only naive ERM is provided as a baseline. How does the approach fair against more specialised algorithms. I don’t expect this novel approach to be state-of-the-art but it would be nice to know where it stands among more modern algorithms.
3. While the authors do compare against [1], I think the paper would benefit from a more in-depth comparison of the two losses. I’m also a bit confused as to why the results of [1] are not reported in Table 1 but only in a separate ablation in Table 2. Could the authors clarify this?

\
\
[1] Yao et al, Pcl: Proxy-based contrastive learning for domain generalization

**Questions:**

1. I think Equation (5) has some typos, shouldn’t the embedding $z_i$ also depend on the environment $e$, i.e. $z_i^e$? If one interprets this equation “literally” you would be summing over the same $z_i$ over and over again.

---

> ### Author Response · Authors · 2023-11-16
> **Response to Reviewer HW2k**
>
> We are grateful for your insightful comments and appreciation of our work. We address each question in detail and provide further clarifications below.
>
> > *Q1: How does Thm 5.1 connect back to the proposed loss function?*
>
> In Thm 5.1, we can see that the upper bound consists of three factors: (1) the optimization error $ε^{1\over 3}$, (2) the Rademacher complexity of the given neural network, and (3) the estimation error which goes down as the number of samples $N$ increases. While (2) and (3) are standard, the critical term, (1) $ε^{1\over 3}$, measures how sample embeddings are aligned with their class prototypes on the hyperspherical space (as we have $\frac{1}{N} \sum_{j=1}^N \boldsymbol{\mu}_{c(j)}^{\top} \mathbf{z}_j \geq 1-\epsilon$), which is optimized by our proposed loss.
>
> Our Theorem implies that improved alignment (as illustrated in Figure 4(b)) leads to smaller upper bound of the intra-class variation $\mathcal{V}^{\text{sup}}$, a key term to upper bound the OOD generalization error by Thm 3.1.
>
> > *Q2: How does our loss compare with more specialized algorithms on CIFAR10-C?*
>
> Thanks for the suggestion! Here we compare our loss (HYPO) with very recent algorithms: EQRM (NeurIPS'22) [3] and SharpDRO (SOTA from CVPR'23) [4], on the CIFAR10-C dataset (Gaussian noise). All three methods are trained on CIFAR-10 using ResNet-18, consistent with the architecture used in our main paper. The results are presented below.
>
> | Method | OOD Acc. (%) |
> |-------|-------|
> | EQRM       | 77.06 |
> | SharpDRO   | 81.61 |
> | HYPO (ours)| 85.21 |
>
> > *Q3: A more in-depth comparison with the PCL loss*
>
> We provide a comparison in the paragraph *Relation to PCL [1]* on page 7. Here we would like to highlight notable distinctions and provide further clarifications:
> - [**Theoretical grounding**] PCL offers no theoretical insights, while our loss HYPO is guided by theory. We provide a formal theoretical justification that our method reduces intra-class variation which is essential to bounding OOD generalization error (Section 5).
> - [**Statistical interpretation via vMF**] While PCL and our loss share high-level intuitions, our loss formulation can be rigorously interpreted as shaping vMF distributions of hyperspherical embeddings (Section 4.2). In contrast, it remains unclear if PCL can be statistically interpreted.
> - [**Strong performance w/o specialized optimization technique SWAD**] Unlike PCL which relies on SWAD [2], a dense and overfit-aware stochastic weight sampling strategy for OOD generalization, HYPO is able to achieve strong performance by simple SGD optimization (Avg Acc 88.0\% without SWAD) and the performance can be further improved with SWAD.
>
> > *Q4: Why the comparison of HYPO vs. PCL is in not included in Table 1?*
>
> Table 1 intends to contrast the effect of different loss functions **under the standard SGD optimization**. We exclude PCL from Table 1 since the performance reported in the original PCL paper (Table 3) is implicitly based on SWAD.
>
> Due to the relevance of two losses and the choice of optimization strategies, we believe it is worthwhile to provide a detailed comparison of HYPO vs. PCL under four optimization strategies: PCL w/o SWAD, HYPO w/o SWAD, PCL w/ SWAD, and PCL w/ SWAD. Therefore, we summarize the results in a separate table (Table 2), as opposed to including them in Table 1 for better clarity. As shown in the table, compared to PCL, HYPO achieves superior performance with (Avg Acc 88.7\% $\rightarrow$ 89\%) and without SWAD (Avg Acc 86.3\% $\rightarrow$ 88\%), which further demonstrates its effectiveness and generality.
>
> Table 2 in the main paper is pasted here for easy reference:
>
> | Model | Art painting | Cartoon | Photo | Sketch | Average Acc. (%)|
> |-----|-----|-----|-----|-----|-----|
> | PCL w/o SWAD  | 88.0 | 78.8 | 98.1 | 80.3 | 86.3 |
> | HYPO w/o SWAD (Ours) | 87.2 | 82.3 | 98.0 | 84.5 | 88.0 |
> | PCL w/ SWAD   | 90.2 | 83.9 | 98.1 | 82.6 | 88.7 |
> | HYPO w/ SWAD (Ours) | 90.5 | 84.6 | 97.7 | 83.2 | 89.0 |
>
> **References**
>
> [1] Yao et al., PCL: Proxy-based Contrastive Learning for Domain Generalization, CVPR 2022.
>
> [2] Cha et al., SWAD: Domain Generalization by Seeking Flat Minima, NeurIPS 2021.
>
> [3] Eastwood et al., Probable domain generalization via quantile risk minimization. NeurIPS 2022.
>
> [4] Huang et al., Robust Generalization against Photon-Limited Corruptions via Worst-Case Sharpness Minimization. CVPR 2023.
>
> > *Q5: Typos*
>
> Thanks for pointing out! We have fixed this typo in the updated manuscript by adding a superscript that explicitly denotes the environment ($\mathbf{z}_i → \mathbf{z}_i^e$) to avoid potential misunderstanding.

---

> > ### Comment · Reviewer_HW2k · 2023-11-20
> > **Reply**
> >
> > I thank the authors for the additional experiments!
> >
> > **Loss function:** I missed the dependence of $\epsilon$ on the alignment, thank you for the clarification. I have a follow-up question; if the insights from this theoretical statement are driving the performance of the algorithm, shouldn't the same loss function *without* the separation term also work? Why do you need the separation term in the first place? How are the two terms competing with each other, i.e. is the introduction of the separation term even negatively affecting the alignment (which is supposed to predict performance). Could the authors clarify this relationship?
> >
> > **CIFAR10-C:** How does PCL perform?
> >
> > **HYPO vs PCL:** Thanks for the clarification!

---

> > > ### Author Response · Authors · 2023-11-21
> > > **Response to Reviewer HW2k (Followup)**
> > >
> > > Thank you for reading our response and following up! We are happy to answer your questions below:
> > >
> > > ### Loss function:
> > >
> > > > Q1. Clarification on the necessity of the separation term
> > >
> > > - **Theoretically**, both low intra-class variation and high inter-class separation are desirable properties for theoretically grounded OOD generalization. This is based on the theoretical framework proposed by Ye et al., which integrates the inter-class separation into the learnability of the OOD generalization problem. In summary, when the learned embeddings exhibit high inter-class separation, the problem becomes learnable. In this context, bounding intra-class variation becomes crucial for reducing the OOD generalization error. We provide a brief summary in Appendix C to discuss the notion of OOD learnability, and would like to refer readers to Ye et al. for an in-depth and formal treatment*.
> > >
> > >
> > >     *The relation between OOD learnability and inter-class separation can be seen in Ye et al., such as the definition of inter-class separation ('informativeness') in Def 3.2 (P5), the necessity of informativeness (P6), and the relation between informativeness and OOD generalization in Thm 4.1 (P7).
> > >
> > >
> > > - **Empirically**, to verify the impact of inter-class separation, we conducted an ablation study in **Appendix H** where we compare the OOD performance of our method (with separation loss) vs. our method (without separation loss). We observe that incorporating separation loss indeed achieves stronger OOD generalization performance.
> > >
> > >
> > > | Method | OOD Acc. (%) |
> > > | -------| ------- |
> > > | HYPO (w/o separation loss) | 87.2|
> > > | HYPO (w separation loss)   | **88.0**|
> > >
> > >
> > >
> > > ### CIFAR10-C:
> > >
> > > > Q2. How does PCL perform?
> > >
> > > We train with PCL loss on CIFAR-10 and evaluate on CIFAR-10-C (same test dataset as ours), the OOD accuracy is 80.68\%.
> > >
> > > | Method | OOD Acc. (%) |
> > > |-------|-------|
> > > | EQRM       | 77.06 |
> > > | SharpDRO   | 81.61 |
> > > | PCL        | 80.68 |
> > > | HYPO (ours)| **85.21** |

---

### Author Response · Authors · 2023-11-16
**General Response**

**Review summary** We sincerely appreciate all three reviewers for their time and effort in providing feedback and suggestions on our work. We are glad that reviewers recognize our paper to be _novel_ (R1, R2), _well-motivated_ (R2), studying _an interesting and important_ (R3), and providing _valuable theoretical contribution_ (R1, R2). We also appreciate that reviewers found our empirical results to be _thorough_, _strong_, and _extensively evaluated across a wide range of tasks_ (R1, R2). All reviewers recognize our paper to be _well-written_ and _easy to follow_ (R1, R2, R3).

**Responses and changes in the manuscript** We have addressed the comments and questions in individual responses to each reviewer. We highlighted the major changes to our manuscript within the document by using blue color text. Here is a summary of the changes we have made following the valuable suggestions from the reviewers:

- Table 1 has been reordered to display the data in ascending order of average accuracy.
- In the introduction, we cited a survey paper and deferred the citations on individual works to Related Works.
- We cited a related work on contrastive metric learning with hyperspherical embeddings.
- We fixed the typo in Equation 5.
- We added a footnote on the distance metric $\rho$ in Thm 5.1.
- We provided empirical verification on the intra-class variation to verify the tightness of the upper bound (Thm 5.1) in practice.
- We added additional comparisons with more specialized algorithms on CIFAR10-C.

---

### Meta-Review · Area_Chair_GzA6 · 2023-12-10

**Metareview:**

This work builds on the theoretical framework introduced by Ye et al. to analyze and motivate prototype-based learning for OOD generalization. It introduces a practical algorithm to achieve improved OOD generalization by aiming to control inter- and intra-class variation. Performance is studied on multiple datasets empirically.

All reviewers agreed that the paper is well-written, provides a practical algorithm, a useful geometric intuition and a thorough empirical study of the algorithm which shows superior performance to other methods in various settings.

Some uncertainty remains after discussions, e.g. how reported $\hat{\epsilon}$ values compare to the loss bound (B) and what the magnitude of $\hat{\epsilon}$ is for other algorithms the method compares to. It would also be desirable to extend the analysis of the algorithm to ground it in a more classical PAC framework. However, most concerns of the reviewers were addressed by the authors in a constructive discussion and hence overall the paper has improved significantly in the discussion phase. Overall, the clear exposition and convincing empirical results warrant acceptance to the conference.

**Justification For Why Not Higher Score:**

Accepting this work is already overruling reviewers and hence, I am not comfortable going any further than poster.

**Justification For Why Not Lower Score:**

See above.

---

### Decision · Program_Chairs · 2024-01-16

Accept (poster)